# Early Holocene Scandinavian foragers on a journey to affluence: Mesolithic fish exploitation, seasonal abundance and storage investigated through strontium isotope ratios by laser ablation (LA-MC-ICP-MS)

**Adam Boethius**[1]*, **Mathilda Kjällquist**[2], **Melanie Kielman-Schmitt**[3], **Torbjörn Ahlström**[1], **Lars Larsson**[1]

**1** Department of Archaeology and Ancient History, Lund University, Lund, Sweden, **2** The Archaeologists, The National Historical Museums of Sweden, Lund, Sweden, **3** Vegacenter, Department of Geosciences, Swedish Museum of Natural History, Stockholm, Sweden

* adam.boethius@ark.lu.se

## Abstract

At Norje Sunnansund, an Early Holocene settlement in southern Sweden, the world's earliest evidence of fermentation has been interpreted as a method of managing long-term and large-scale food surplus. While an advanced fishery is suggested by the number of recovered fish bones, until now it has not been possible to identify the origin of the fish, or whether and how their seasonal migration was exploited. We analysed strontium isotope ratios ($^{87}Sr/^{86}Sr$) in 16 cyprinid and 8 pike teeth, which were recovered at the site, both from within the fermentation pit and from different areas outside of it, by using laser ablation multi-collector inductively coupled plasma mass spectrometry. Our investigation indicates three different regions of origin for the fish at the site. We find that the most commonly fermented fish, cyprinids (roach), were caught in the autumn during their seasonal migration from the Baltic Sea to the sheltered stream and lake next to the site. This is in contrast to the cyprinids from other areas of the site, which were caught when migrating from nearby estuaries and the Baltic Sea coast during late spring. The pikes from the fermentation pit were caught in the autumn as by-catch to the mainly targeted roach while moving from the nearby Baltic Sea coast. Lastly, the pikes from outside the fermentation pit were likely caught as they migrated from nearby waters in sedimentary bedrock areas to the south of the site, to spawn in early spring. Combined, these data suggest an advanced fishery with the ability to combine optimal use of seasonal fish abundance at different times of the year. Our results offer insights into the practice of delayed-return consumption patterns, provide a more complete view of the storage system used, and increase our understanding of Early Holocene sedentism among northern hunter-fisher-gatherers. By applying advanced strontium isotope analyses to archaeological material integrated into an ecological setting, we present a methodology that can be used elsewhere to enhance our understanding of the otherwise elusive indications of storage practices and fish exploitation patterns among ancient foraging societies.

**Data Availability Statement:** All relevant data are within the manuscript and its Supporting information files.

**Funding:** We are grateful for the financial support from: AB, The Swedish Research Council, VR-2019-02975, https://www.vr.se/. AB, Birgit och Gad Rausings Stiftelse för Humanistisk forskning, https://www.rausing.org/. The funders had no role in study design, data collection and analysis, decision to publish, or preparation of the manuscript.

**Competing interests:** The authors have declared that no competing interests exist.

# Introduction

Hunter-fisher-gatherer subsistence strategies are often related to optimal foraging theory [1, 2]. This generally implies a close connection with mobility patterns, with the assumption that once a particular part of the landscape begins to run low on resources, i.e. when search and pursuit costs increase above the cost of moving to a different area (see the marginal value theorem [3]), people move from that area, as a risk-reducing strategy [4]. However, among societies that are largely dependent on reliable aquatic resources for sustenance, mobility is often less important [5, 6]. For these groups of people, favourable locations within the landscape have increased importance [7], e.g. where fish are available all year round and where it is possible to exploit massively increased numbers of fish during limited periods (i.e. during fish migrations or spawning activities). The capitalization of annual fish runs, migrations and congregations have for millennia been important for fish-dependent societies [8], and even today these yearly patterns are exploited by both professional and recreational fishermen.

Among foraging societies, subsistence strategies based on aquatic resources are often considered to be a factor in increasing population densities [9]; in particular, the ability to exploit migrating fish on a large scale, e.g. as among northwest coast American societies prior to European contact, is often viewed as a driver of affluence [10]. This, in turn, has other societal implications, such as a territorialization of the landscape, because the locations where the fish runs can be optimally exploited increase in importance and become the focus of intergroup competition, especially when resources occur abundantly during limited periods of the year but are scarce during other periods [11]. The potential to acquire vast surpluses is considered to be one of the prerequisites for generating social inequalities and ranked hunter-fisher-gathering societies, when combined with the ability to store the surplus for later use [12, 13], i.e. the practice of delayed-return subsistence strategies [14]. Consequently, the ability to use seasonal fish abundance optimally can be seen as a stage in the development of complex (non-egalitarian) societies, especially when other criteria are met, such as population pressure, temporal and spatial limited resource abundance and competition over the rights to those locations, labour control and tribal warfare [13, 15–18]. However, when no ethnographic accounts are available and when taphonomic processes have erased much of the organic record, the potential for understanding ancient hunter-fisher-gathering societies, their level of complexity and their utilization of fish migrations, is significantly diminished, and many criteria for understanding forager complexity remain elusive [19].

In some areas, human utilization of fish seasonal migrations can be studied through standard zooarchaeological analyses, e.g. by identifying a large number of salmon bones from a particular site [20–22], or by analysing $\delta^{13}C$ ratios to determine whether the fish had spent their entire life in freshwater or been caught during spawning migrations between saltwater and freshwater (anadromous and catadromous species or subspecies) [23]. However, understanding past human use of fish migration and spawning that takes place wholly within a freshwater environment (potamodromy) is difficult. Because of the lack of reliable methodologies, potamodromous spawning migrations have been the focus of ecological rather than archaeological studies. Furthermore, even in ecological studies, it is only recently, as a result of new methods, such as environmental DNA, that large-scale analyses have become more manageable [24, 25].

The Mesolithic period is known for its diversification of subsistence strategies and resources [26, 27], and a large species diversity is often encountered on Mesolithic sites that have favourable preservation of organic remains [28–34]. In recent years, the importance of fish in Scandinavian Mesolithic societies has been highlighted [19, 35–39], with an inferred important role in a suggested regionalization and territorialization of the Scandinavian Mesolithic landscape

[19, 40, 41]. New collations of European fish bone records show that the quantity of identified fish bones from Sweden, Denmark and Norway combined amounts to roughly two-thirds of the entire Mesolithic European fish bone assemblage [42], again highlighting the importance of fish and its impact on early northern foraging communities.

There are many reasons why fish appear to have been so important among northern Mesolithic foragers. Some of them are methodological, such as large efforts of fine-mesh water sieving during excavation and favourable preservation conditions, as well as modern land exploitation and more intense land development in certain areas, in combination with a somewhat stronger research tradition in Mesolithic archaeology than in other parts of the world. However, even with a methodological bias, the fish bone record from northern Europe highlights the importance of fish to the Mesolithic people, even more so when other evidence is considered, such as fishing nets, traps and weirs [43–46], hooks, netting needles and sinkers [47–50], and boats, paddles and other indirect indications of watercraft [51–56]. This strong relationship with aquatic exploitation has led to the suggestion that humans exploited their local aquatic environments to such a large degree that it had already affected aquatic species composition and abundance in the Early Holocene [41].

## Norje Sunnansund

Until recently, most of the evidence for Scandinavian Mesolithic fish exploitation has come from Late Mesolithic Ertebølle contexts [57–60], suggesting that heavy marine resource exploitation drove a human population increase in southern Scandinavia in the centuries before agricultural introduction [61]. There has been scant evidence of fish exploitation from earlier periods [19], but during the last decade, new data have emerged from e.g. the excavation of the boreal site Norje Sunnansund, dated to around 9600–8600 calibrated (cal) years before present (BP). Norje Sunnansund is located in southeastern Sweden, in Blekinge county, and was, at the time of occupation, located on the shores of the ancient Lake Vesan next to an outlet to the Baltic Sea, which was located around 2 km from the site (Fig 1). The site was located in an ecotone setting, with three different water bodies, surrounded by pine and hazel-dominated forest, with a low mountain ridge to the west adding to environmental and geological diversity. The site was occupied during two separate phases, stratigraphically recognized as two different archaeological layers deposited on land and one archaeological layer deposited in the shallow waters outside the shores of ancient Lake Vesan (S1 Fig in S1 File). Furthermore, a pit used for fermenting large amounts of fish had been dug close to the water edge within the oldest land based layer and approximately 10–20 cm down into the underlying clay [62].

All evidence (radiocarbon dating, stratigraphic and contextual information) indicate that the fermentation pit was contemporaneous with the human use of the other parts of the site (thus being comparable in terms of sedimentary horizons, geological substrates and bioavailable $^{87}Sr/^{86}Sr$ ranges). The site displays all year round human occupation, with an increasing number of seasonality indicators between September to May [29], suggesting more limited use of the site during summer. The site is the earliest identified winter season / all-year-round settlement from southern Scandinavia.

"Modern" Lake Vesan was systematically drained, starting at the beginning of the 19th century until the lake was completely dried out in the 1930s, to create arable farmland [63]. The site Norje Sunnansund was found during surveys in 2009 and excavated in 2011, prior to a road construction cutting though the site [63]. The site is covered with a 5–120 cm layer of gyttja (depending on which part of the site) deposited from the sea that transgressed the site shortly after the Mesolithic occupation (cf. e.g. [63, 64] for details of the site formation). The inundated nature of the site and the thick layer of gyttja in the area of good organic

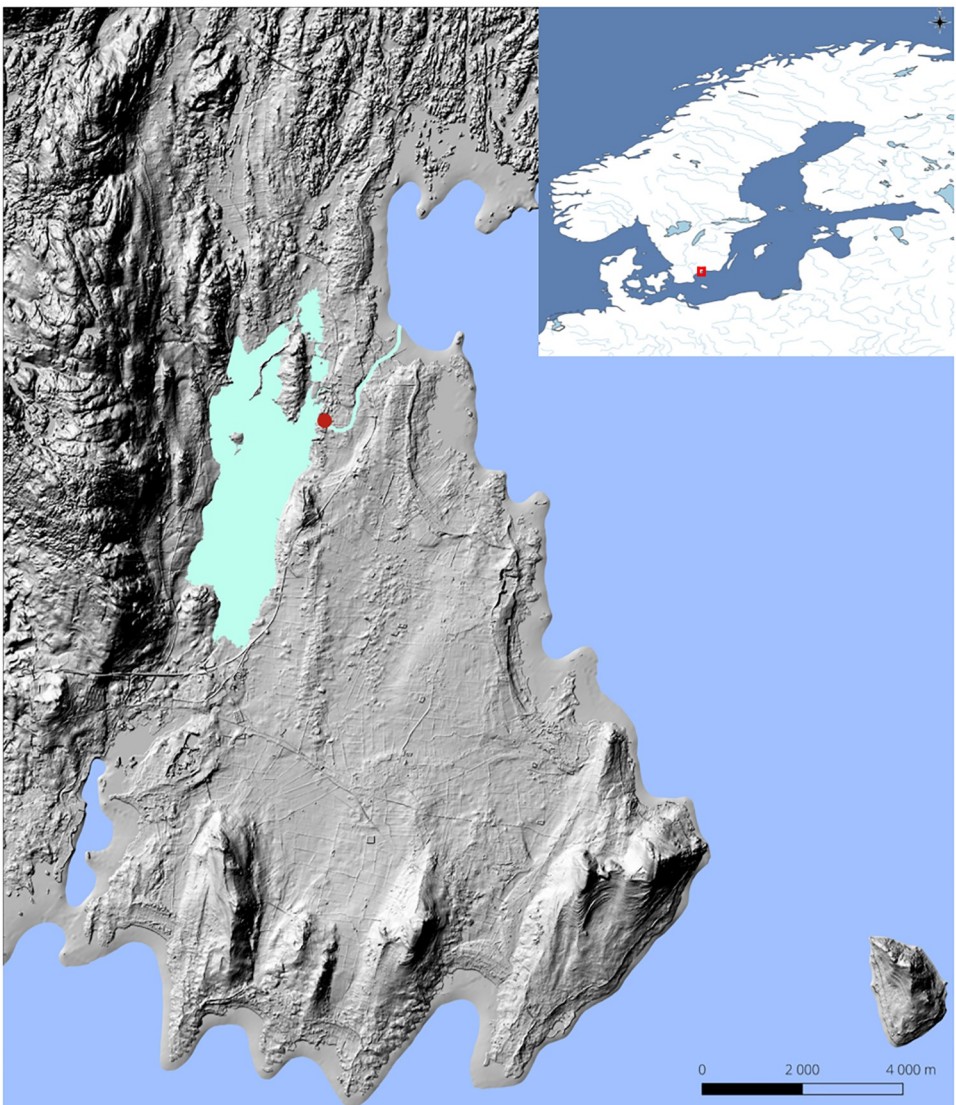

**Fig 1. Map of the Norje Sunnansund area and site (red dot) during the time of occupation around 9000 cal BP.**
World Map generated with QGIS 3.10 using the Natural Earth data set. The local map is based on a terrain model at a
5-m resolution, LIDAR data and topographic information from the Swedish Land Survey, Lantmäteriet i2012/892,
Swedish Geological Survey and Iowtopo2 [65]. Map by Nils-Olof Svensson, Kristianstad University. The local map is
modified from original previously published in [63] and is, with permission from Blekinge Museum, freely available
through CC by 4.0 license.

preservation in combination with a rapid transgression following the abandonment of the
site have prevented any later intrusions from burrowing animals. Consequently, the entire
faunal record found at the excavation of the site is considered to be contemporaneous with
the human occupation of Norje Sunnansund.

As a result of both favourable preservation conditions and water sieving of all excavated
soil, the site has yielded large quantities of fish bones. To date, 16,180 fragments have been
identified to species or family level [29], yet only about 13% of the recovered fish bones have
been analysed, and less than 10% of the original site has been excavated. Taking this into con-
sideration, estimations of the original fish catches indicate that hundreds of tons of fish were

probably caught at the site [66]. These quantifications show that large-scale fisheries existed in Scandinavia millennia before the emergence of the Ertebølle period. Furthermore, the fermentation pit from Norje Sunnansund is the earliest known evidence of fermentation, and large amounts of fish were fermented in a pit surrounded by post- and stakeholes (Fig 2) and stored for later use [62]. The ability to store large amounts of fish for later consumption, combined with the large quantity of fish that was caught and year-round site occupation [29], suggests a managed fishery with the ability to practise delayed-return subsistence strategies and utilize the fish during different seasons.

Diverse fishing practices are indicated by the spatial patterns of fish species diversity and abundance from different areas of the site [62, 63] (Fig 3). The species distribution patterns are especially evident when comparing the fish fermentation pit with other areas of the site. In the pit the fish bone assemblage is dominated by cyprinids, in particular roach (*Rutilus rutilus*), representing 79% of the assemblage compared with 22–56% in other areas of the site (Fig 3). This suggests that other fish species were targeted when different methods of processing, storing or eating the fish, other than fermentation, were to be applied.

The large proportion of roach in the pit, combined with the practice of fermentation to facilitate long-term and large-scale storage, suggests that the fish had been caught to generate a surplus. There are indications that the fermentation process was a recurring event [29, 62], and

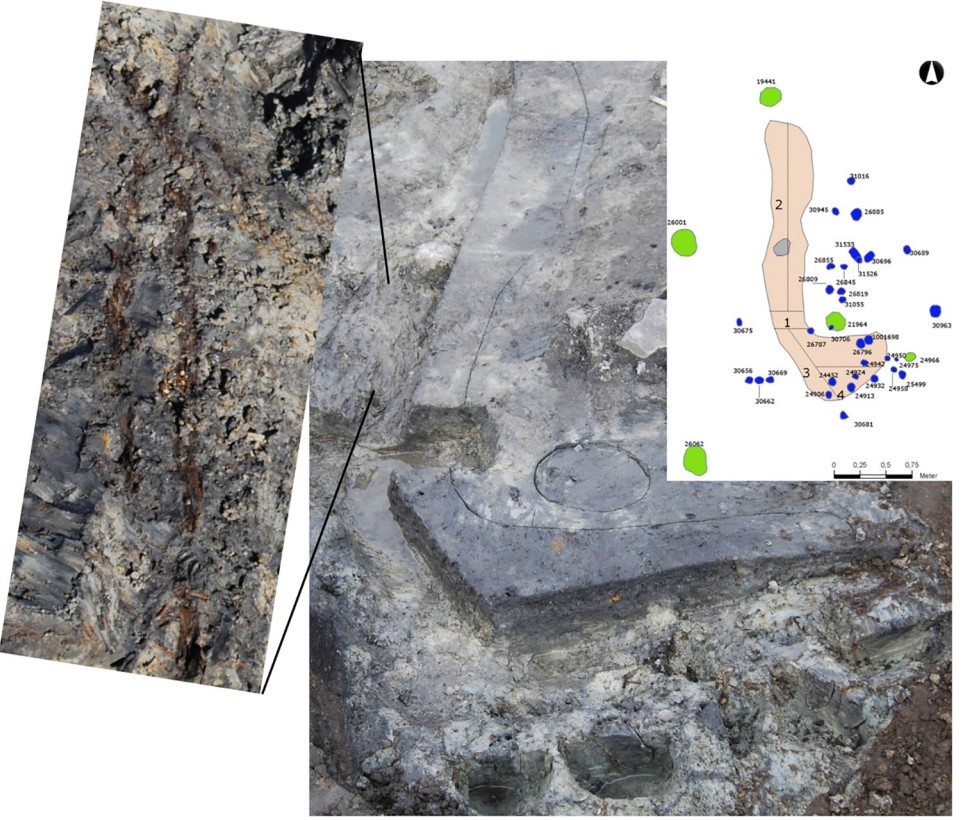

**Fig 2. Picture of the fermentation pit surrounded by larger postholes (green) and smaller stakeholes (blue) with plant fibres lining the clay wall (left).** The picture was taken when the upper part had already been removed and is shown as an elongated pit dug down into the underlying clay. Photo: Adam Boethius. Images originally published in [63] and are, with permission from Blekinge Museum, freely available through CC by 4.0 license.

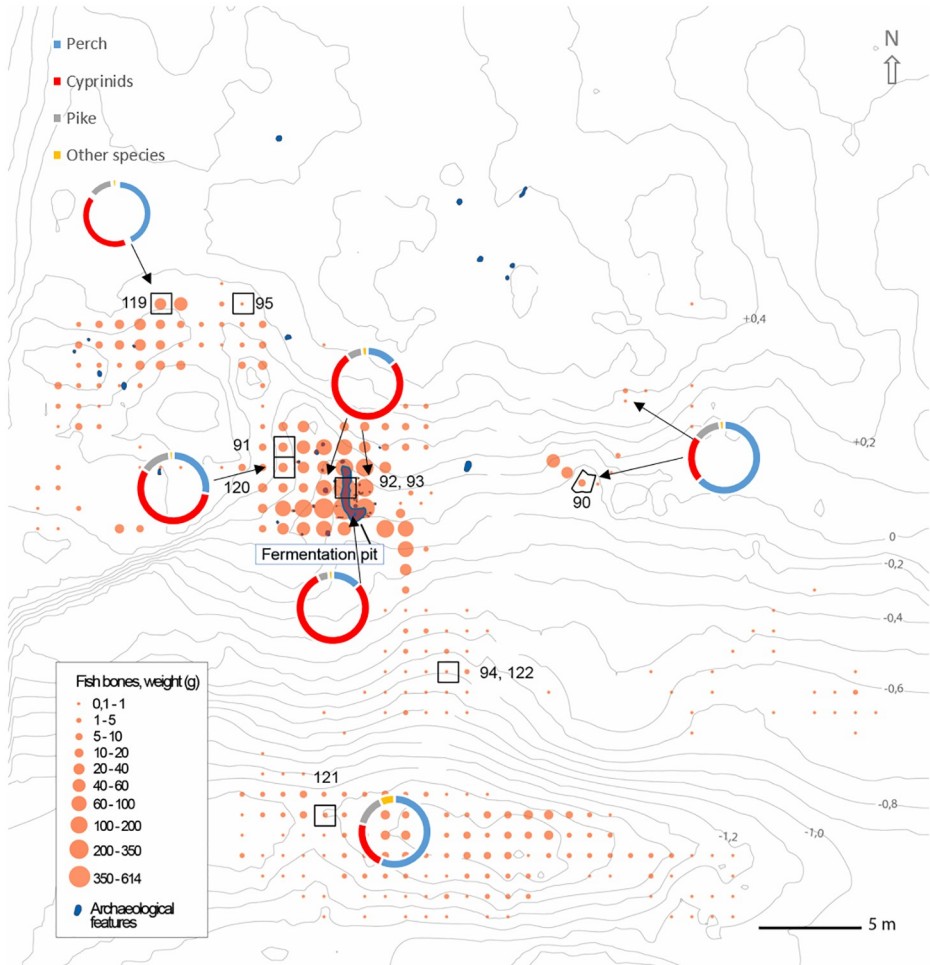

**Fig 3. Map of the site layout.** Showing the location of the fermentation pit, the amount of recovered fish bones (the size of the orange dots reflects the weight of the recovered fish bones from each square) and the fish species data from the different areas of the site (north-west, west, central area outside the pit, east, fermentation pit and south). The find locations of the ablated teeth recovered outside the fermentation pit are shown with their number and a black square as follows: cyprinids: 90, 91, 95, 119, 120, 121, 122 and pike: 92, 93, 94. The teeth recovered from within the fermentation pit are not shown on the map but are as follows: cyprinids: 76, 77, 78, 79, 80, 81, 82, 83, 84 and pike: 85, 86, 87, 88, 89. Note that pike 92 & 93 are located in a cultural layer deposited stratigraphically overlying the fermentation pit and are not from the pit (cf. S1 Fig in S1 File and [63] for detailed information on stratigraphic relations).

it has been hypothesized that the fish fermentation was facilitated by people using fish spawning periods to extract large quantities of fish [19]. The fermented fish would enable long-term storage that, in turn, would enable a more sedentary lifestyle, because the stored product could be used during times of scarcity and therefore reduce the need to move into new areas as local resources temporally declined (cf. e.g. [4, 9]).

Until now, it has not been possible to investigate the hypothesis that fish spawning was used to create a surplus, or whether the fermented fish is an example of intensive use of a productive lake. The use of local fish, even if caught in abundance, would imply that the fish was fermented on a smaller scale and possibly solely for immediate culinary reasons. In contrast, making full use of fish spawning and migration implies very large catches were made during limited periods of the year and that the fermentation process was used primarily to

facilitate large-scale and long-term storage, which in turn could function as a driver for a more populous, sedentary and territorial society.

## Prerequisites for studying provenance and mobility using strontium isotope ratios

Mobility and provenance studies are an integral part in archaeological research and have traditionally used archaeological methods, such as cultural specific objects and technological traits but have since the 1980s also involved the analysis of $^{87}Sr/^{86}Sr$ isotope ratios, cf. e.g. [67, 68]. The principle of using Sr isotopes in mobility and provenance studies lies within the bedrock of a region where e.g. an older granitic bedrock has a higher $^{87}Sr/^{86}Sr$ ratio than a young basaltic bedrock [69]. The stable isotope $^{87}Sr$ forms through the emission of negative β-particles from Rubidium-87 ($^{87}Rb$) [70]; whereby, the $^{87}Sr/^{86}Sr$ ratio in the bedrock is dependant on both the original content of rubidium and the age since rock formation. Because the half time of $^{87}Rb$ is very long (ca. 49 billion years) no additional Rb decay needs to be accounted for in archaeological studies. Furthermore, the $^{87}Sr/^{86}Sr$ ratio remain unaltered as it is transferred from the bedrock to the top-soil and further into water and plants in a particular area [71]. As animals (including humans) feed and drink they absorb the bioavailable $^{87}Sr/^{86}Sr$ values (range of available $^{87}Sr/^{86}Sr$ ratios at a specific location) into their bodies where it is stored. In soft tissue, hair and bone, the Sr-values are continuously replaced throughout life, when the body remodels. However, some elements of the body, such as teeth and fish otoliths, remain unaltered, once they have formed; whereby, the Sr-ratios in tooth enamel represents the location during the formation of a particular tooth. Because of these basic principles, the bioavailable baseline in the landscape may be correlated to specific $^{87}Sr/^{86}Sr$ tooth data to establish a probable region of origin. Or, when more high-resolution methods are applied, to establish how an animal moved during the formation of a tooth or an otolith [72–74]. For more information of how strontium is incorporated into a body see e.g. [75] and for further information of bioavailable strontium and discussions of its use in archaeological research see e.g. [76–78]. For information of its implementation in fish ecological research (primarily done on otoliths) and how fish migrations and mobility is integrated with environmental isotopic variations across space and time, i.e. isoscapes [79, 80], see e.g. [81–84].

## Fish tooth development

Cyprinid fish are polyphyodont [85], which means that they continuously replace their dentition throughout life. However, their tooth development is similar to that of mammals [86], so it is possible to study mobility patterns based on the formation of a specific tooth, even though it is not possible to connect the data to a specific age of that fish. Cyprinid teeth are located in the throat and not in the mouth of the fish; these pharyngeal teeth develop in different stages, referred to as tooth generations [87]. The first tooth generation is not attached to the pharyngeal bone, while later generations are attached to the bone before being resorbed as new tooth generations, located at the same position on the bone, develop. In roach, the adult dentition appears around 45–50 days after hatching [88], so the Sr data do not reflect the first 7 weeks of life.

   Northern pike (*Esox lucius*), hereafter referred to as pike, is also a polyphyodont species [89]. The development of pike teeth has been described in detail by Herold [90] and, regarding the sequence of enamel formation, again corresponds with the development of mammal teeth. Thus, while unrelated to the age of the fish, a pike tooth holds, similar to cyprinids, information on the origin and mobility of the fish within the timespan of the development of each tooth. The exact duration and timing of pike teeth development has not been thoroughly researched.

However, pike has, because the adult dentition is characterized by numerous teeth of different size and shape throughout the oropharyngeal region, recently been identified as "a powerful model system for studying tooth development" [91]; whereby, more exact data on fish tooth development can hopefully be produced over the coming years.

## Materials

### Fish tooth specimens

To investigate the provenance and mobility of the fish from the site, from both within the fermentation pit and from outside of it, and to study the possible human utilization of seasonal fish spawning and congregation, 24 fish teeth from Norje Sunnansund were sampled for analysis. The samples included 8 teeth from pike and 16 teeth from cyprinids (Cyprinidae). The cyprinid samples included 11 teeth from roach, 1 tooth from bream (*Abramis brama*), 1 tooth from rudd (*Scardinius erythrophthalmus*), 1 tooth from dace (*Leuciscus* leuciscus) and two teeth from unidentified cyprinids. Zooarchaeological analyses of the fish teeth were carried out using the comparative collections at the Department of Archaeology and Ancient History, Lund University, Sweden, and at The Archaeologists, National Historical Museums, Sweden.

The selected fish teeth (S2-S25 Figs in S1 File) were sampled from different areas of the site (Fig 3), where variations in species distribution indicated different types of fishing strategies/methods or different ways of preparing fish caught during different seasons [62, 63]. This sampling strategy meant that fish recovered both within and outside the fermentation pit could be compared and it limited the likelihood of the same individual being sampled more than once. The risk of double sampling was also deemed unlikely due to the very large number of recovered fish teeth from a large number of identified fish individuals and based on estimations suggesting the possibility of hundreds of tons of originally caught fish at the site [66].

After selection, the fish teeth were taken to the Vegacenter at the Museum of Natural History, Stockholm, Sweden, for $^{87}Sr/^{86}Sr$ strontium isotope ratio analyses using laser ablation multi-collector inductively coupled plasma mass spectrometry (LA-MC-ICP-MS).

### Baseline data

To contextualize the $^{87}Sr/^{86}Sr$ strontium isotope ratios obtained from the ablated fish teeth and to establish a bioavailable $^{87}Sr/^{86}Sr$ strontium isotope ratio map for southern Sweden, previously existing data from both archaeological and modern fauna [92] and data from previously published water analyses [72, 93] were used.

The local terrestrial Sr baseline at Norje Sunnansund have previously been established by bulk $^{87}Sr/^{86}Sr$ isotope analyses on three field vole teeth (*Microtus agrestis*). However, because fish is studied here, the aquatic baseline at Norje Sunnansund needed investigation. Unfortunately, modern $^{87}Sr/^{86}Sr$ ratios in the local area have been subjected to agricultural and lake liming and are likely not representative for Mesolithic Sr ratios. Thereby, data from modern water, floral or faunal sources was avoided. To remedy the problems of using modern sources, the baseline studies were expanded to include archaeological samples of a local stationary species with an aquatic orientation. Thus, 21 laser ablation analyses on seven water vole teeth (*Arvicola amphibious)* were conducted, of which one struck dentine and was excluded. To further delimit and enable estimations of the original $^{87}Sr/^{86}Sr$ ratio within the ancient Lake Vesan, unlimed water sources were sampled from the tributary system along the water catchment route leading down to the Vesan valley. These include two nearby lakes (Vitavatten and Stora Svartsjön), one natural spring (Hålabäck) and one remote lake (Bredagylet). Two other natural springs (Barnakälla and Skönabäckskällan) were also sampled to provide bioavailable baseline data from nearby areas with a younger geological bedrock. All

sampled water sources derive from less than 30 km from the site, but from areas with different underlying bedrock and sedimentary horizon. The sampled water sources have not been subjected to any documented liming, which most other water systems have been in the area (cf. the Swedish National Lime database map).

## Methods

### Ethics

The archaeological tooth specimens included in this study (7 from water voles, 16 from cyprinids and 8 from pikes) were excavated in 2011, prior to the construction of the E22 highway cutting through the area of the Norje Sunnansund site. The excavation was done according to Swedish legislation, with permission from the County Administrative Board of Blekinge, Sweden (reference number 431-1581-11, 431-2149-11). All recovered archaeological remains are temporally stored at the Archaeologists, National Historical Museums, Lund, Sweden, but will be transferred to Blekinge Museum for permanent deposition. The laser ablations have caused minimal damage to the teeth, which is included in the overall permission to excavate, analyse and interpret the archaeological remains.

### Baseline water $^{87}Sr/^{86}Sr$ analyses

Six water samples, to establish the available $^{87}Sr/^{86}Sr$ ratios in the areas surrounding Norje Sunnansund and to enable estimations of the original $^{87}Sr/^{86}Sr$ ratios of ancient Lake Vesan, were analysed on a Thermo Scientific TRITON TIMS using a load of purified sample mixed with tantalum activator on a single rhenium filament. Two hundred 8 second integrations were recorded in multi-collector static mode, applying rotating gain compensation. The measured $^{87}Sr$ intensities were corrected for Rb interference using $^{87}Rb/^{85}Rb = 0.38600$ and ratios were reduced using the exponential fractionation law and $^{88}Sr/^{86}Sr = 8.375209$. The external precision for $^{87}Sr/^{86}Sr$ as judged from running 987 standards was 18 ppm. Accuracy correction was applied, the $^{87}Sr/^{86}Sr$ ratio for the NBS 987 standard was in August 0.710210±13 (n = 12) and in October 0.710244±12 (n = 7), which was normalized to a NBS987 $^{87}Sr/^{86}Sr$ ratio of 0.710248 following [94]. All water analyses were conducted at the Thermal Ionization Mass Spectrometer (TIMS) at the Museum of Natural History, Stockholm, Sweden.

### Laser ablation sample preparation

Before laser ablation, all teeth were gently cleaned in deionized water with a toothbrush. Following cleaning, the teeth were allowed to air dry at room temperature. They were then mounted and fixed, without further treatment or physical manipulation, in a sample cell on a movable mounting table. While in the mounted position, the teeth were wiped with ethanol to remove any superficial stains derived from handling.

### Laser ablation

$^{87}Sr/^{86}Sr$ ratios from the fish teeth were obtained by taking measurements from the enamel surface of each tooth after the tooth had been mounted in a NWR193 excimer laser ablation system (Electro Scientific Industries, Portland, OR, USA) coupled to a Nu Plasma II multi-collector ICP mass spectrometer (Nu Instruments Ltd, Wrexham, UK). All measurements were taken at the Vegacenter laboratory, on four separate occasions (November 2019 and February 2020 for the fish teeth and September-October 2020 for the water vole teeth (*Arvicola amphibious*)). The instrument operating conditions for these occasions are listed in S1-S8 Tables in S1 File and the isotope data for each ablated tooth are given in S9-S39 Tables in S1 File.

All ablations were made on the outer surface of the tooth enamel, following the development of the tooth, i.e. starting at the tip of the tooth and moving down towards the enamel–dentine junction. To remove additional potential surface contamination, prior to each ablation each tooth was pre-ablated using a laser spot size of 150 μm. Parallel tracks ca 400 μm in length using a 130 μm spot size were then ablated for Sr analysis, on the pre-ablated enamel surface (see Fig 4). The number of ablation lines per tooth ranged from 5 to 17 for cyprinids, and from 10 to 23 for pike, depending on the size of the tooth.

The ablations targeted the enamel; however, in some cases, because of variations in enamel cover on the fish teeth and an often thin enamel layer, some ablations included dentine. This was assessed visually by studying close-up photographs of each ablated tooth (S2-S25 Figs in S1 File) and through divergence in $^{87}$Rb/$^{86}$Sr ratios, as this ratio often changes when dentine or bone is ablated (cf. S9-S32 Tables in S1 File). When this occurred, it was noted and the measurements striking dentine were excluded from the interpretations. This is important because bone and dentine have a lower density and more porous structure than enamel [75] and can be more easily contaminated [95]. Furthermore, the LA-MC-ICP-MS method has been developed for measuring tooth enamel and the reference standards provide verification for enamel-based measurements. This is mainly because of the higher organic content of dentine, which can introduce additional interference that cannot be securely accommodated. Consequently, while, if uncontaminated, the enamel-dentine mix could provide additional information, because of the problems of verifying any potential fractionation and interference for dentine data, it cannot be ruled out that observed patterns stemmed from analytical uncertainties.

Enamel can also be contaminated by its depositional soil environment. To ensure that the enamel of the teeth was not subjected to soil contamination we preablated each ablation area (thus discarding the outer few μm of the enamel, cf. instrumental settings S1-S4 Tables in S1 File). The results indicate that we do not see any increases in either rare earth elements (REE)

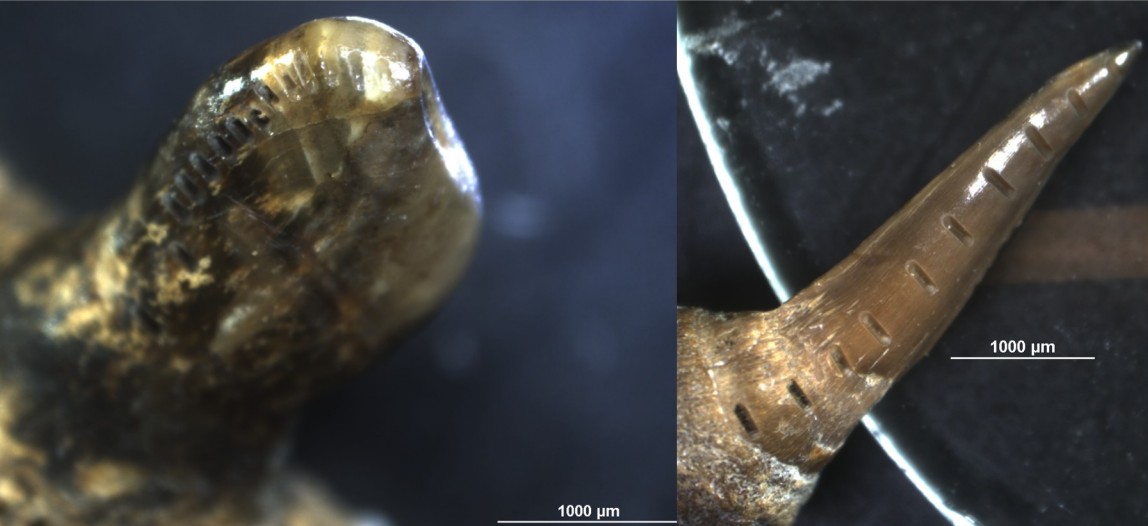

**Fig 4. Photographs of fish teeth showing the marks left by the laser ablation.** Left: Roach tooth 82, with the first ablation at the tip of the tooth and subsequent ablations made in order towards the pharyngeal bone. Note that only the first five ablations are made on just enamel and the later nine ablations are made on a mix of both dentine and enamel, or only dentine. Right: Pike tooth 89, with the first ablation at the tip of the tooth and subsequent ablations made in order towards the dental bone. Note that only the first eight ablations are made on just enamel and the last two ablations are made on a mix of both dentine and enamel or only dentine. All ablations including dentine were excluded.

or Rb, compared to the standards, which would indicate contamination. The spread and variation of the fish enamel measurements offer further indications that the enamel has not been contaminated by its depositional soil environment.

Possible isobaric interferences were corrected by subtracting a gas blank ([84]Kr) and by peak stripping (e.g. doubly charged REE, Ca-dimers/argides, [87]Rb) cf. [72]. All corrections were applied online, which resulted in interference-free $^{87}Sr/^{86}Sr$ ratios for each data point (S9-S39 Tables in S1 File).

A polyatomic interference on *m/z* 87 has also been reported [96] and is described as (Ca/Ar)$^{31}$P$^{16}$O$^+$. This interference can introduce a significant offset in $^{87}Sr/^{86}Sr$, especially for samples with low Sr concentrations. It cannot be corrected online but has to be reduced by a thorough low oxide tuning of the gases [97]. We demonstrate that accurate $^{87}Sr/^{86}Sr$ ratios are obtained even at low Sr concentrations (tested down to $C_{Sr}$ ~ 7 ppm) under optimal tuning conditions (S33 Fig in S1 File). See also [72] for a previous application of the method.

To verify that all interferences had been successfully removed, two separate standards (S5-S8 Tables in S1 File), a spine from a velvet belly lantern shark (*Etmopterus spinax*) (primary reference material (RM)) and a tooth from a European hare (*Lepus europaeus*) (secondary RM), with known Sr concentrations and ratios, were repeatedly analysed throughout the ablation sessions. The analyses on the shark spine yielded a $^{87}Sr/^{86}Sr$ ratio of 070918 ± 0.00028 (2SD, n = 204, November 2019), 0.70919 ± 0.00032 (2SD, n = 291, February 2020), 0.70918 ± 0.00036 (2SD, n = 19, September 2020) and 0.70918 ± 0.00034 (2SD, n = 33, October 2020) which agrees well with the value of seawater of 0.709179 ± 0.000002 as established by [98]. The hare tooth yielded a $^{87}Sr/^{86}Sr$ ratio of 0.71009 ± 0.00026 (2SD, n = 125, November 2019), 0.71009 ± 0.00027 (2SD, n = 68, February 2020), 0.70991 ± 0.00020 (2SD, n = 4, September 2020) and 0.70999 ± 0.00030 (2SD, n = 7, October 2020). This agrees well with the value determined by thermal ionization mass spectrometry (0.709988 ± 0.000015; Finnigan Triton, Thermo Scientific, Waltham, MA, USA) at the Swedish Museum of Natural History.

## Results

### Baseline data

Bulk stable analyses of three rodent teeth, from field voles recovered from Norje Sunnansund, have already been performed and published [92]. These values represent a local terrestrial Sr baseline for the site and range from 0.7166 to 0.7180, with a mean of 0.7172 ± 0.0008 (1 SD). In addition, stable Sr isotope data from both archaeological and modern fauna from different locations around southern Scandinavia have been published [92], along with baseline data from water sampling stations along the coastline of the Baltic Sea [72] (Fig 5). In general, these data sets agree and present an increase in Sr isotope ratio values with increasing latitudes. The detailed picture is more complex, as the Sr signals correspond with the bedrock, and large variations in geological characteristics mean large variations in bioavailable Sr ratios, even on a small scale and when corresponding with local geology.

**Water source analyses.** At the area in the direct vicinity of Norje Sunnansund, estimating an accurate baseline is more complex as the local geological characteristics are highly diverse (Fig 6) and because the former lake has been drained to create arable farmland (S34 Fig in S1 File). Furthermore, the entire farmed valley of the former lakebed has since been subjected to agricultural lime and fertilization along with most of the lakes and rivers in the tributary system (apart from the selected lakes sampled here to investigate the aquatic baseline). It has been suggested that the groundwater, streams and minor water bodies do not reflect the original $^{87}Sr/^{86}Sr$ ratios in areas where agricultural liming has occurred [99]. Counter arguments have been added to suggest that agricultural liming of soils does not affect the groundwater and

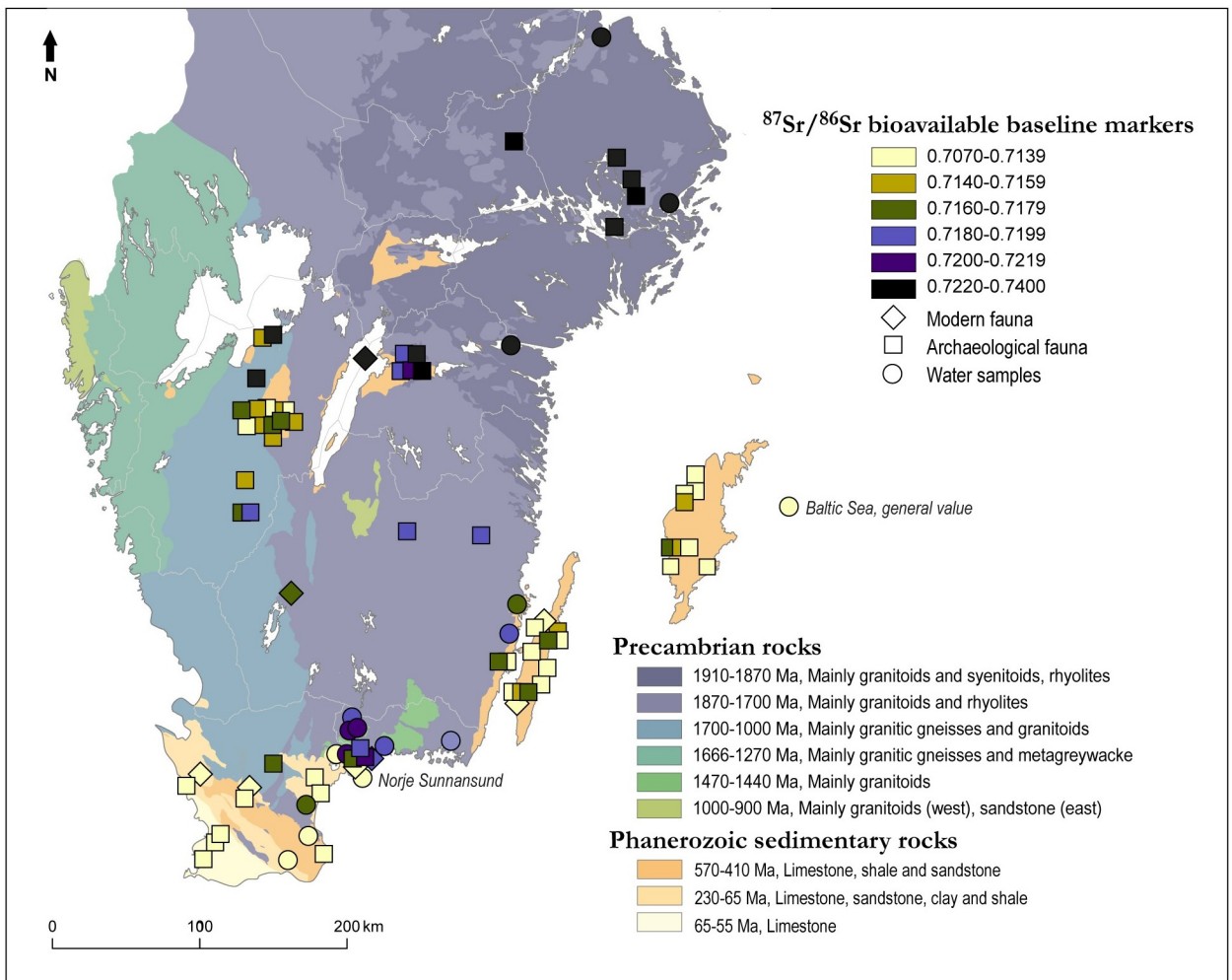

**Fig 5. Bioavailable $^{87}Sr/^{86}Sr$ baseline map of southern Sweden.** Data from [72, 92, 93] and this study. Markers show a range of values from sites where more than one data point is available. Background map based on a simplified bedrock map from Bedrock map 1:1 M, Geological Survey of Sweden.

instead only affect the topsoil [100]. However, here, the aquatic baseline of a former lake is sought and almost all of the lakes and rivers along the water catchment route in the tributary system leading down to the valley where the lake was previously located have been subjected to extensive liming. Thereby, any data obtained from modern local sources (water, plants or rodents) cannot safely be used in estimating the baseline of ancient Lake Vesan.

To remedy this situation we sampled six separate, unlimed, water sources, which gave reference and delimiting $^{87}Sr/^{86}Sr$ ratios from non-local, but nearby, areas around Norje Sunnansund (Fig 6 and S40 Table in S1 File). Four of the water sources follow the water catchment route that once delivered water to Lake Vesan (Fig 6), thus representing the origin of the original ancient Lake Vesan $^{87}Sr/^{86}Sr$ baseline. The two remaining water sources represent the $^{87}Sr/^{86}Sr$ baseline of the close by sedimentary bedrock areas and are, given the high degree of local liming, the best available sources to represent a Mesolithic $^{87}Sr/^{86}Sr$ aquatic baseline in those areas.

**Water vole analyses.** The water in Lake Vesan originated from geologically much older bedrock than the sedimentary bedrock surrounding the former lake. Thereby, the aquatic

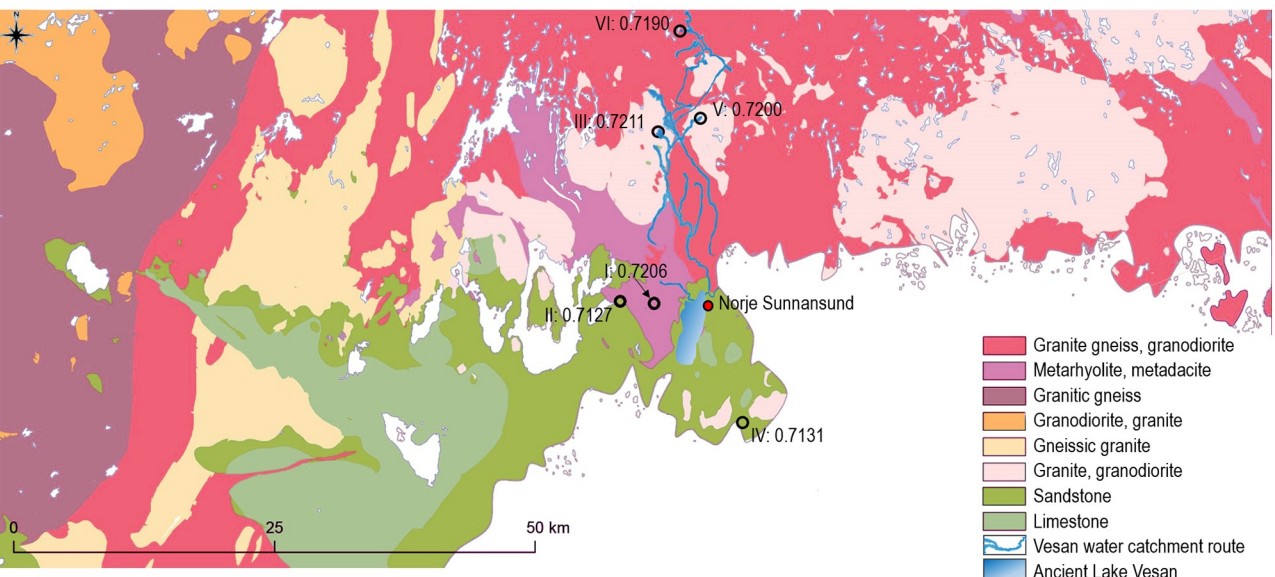

**Fig 6. Bedrock map showing the diverse geological landscape of the Norje Sunnansund area.** The map is based on data from Bedrock map 1:1 M, Geological Survey of Sweden. Bedrock listed according to age showing the older Precambrian rocks: Granite gneiss, granodiorite (1800–1700 Ma); Metarhyolite, metadacite (1770–1750 Ma); Granitic gneiss (1700–1600 Ma); Granodiorite, granite (1600–1500 Ma); Gneissic granite (1500–1200 Ma); Granite, granodiorite (1470–1440 Ma) followed by the younger Phanerozoic sedimentary rocks: Sandstone (205–66 Ma) and Limestone (99–66 Ma). The local $^{87}Sr/^{86}Sr$ baseline for ancient Lake Vesan is complicated to establish due to the drainage of the modern lake, to create arable land and extensive liming of both soils and water sources within the whole area of Blekinge county. The six water sample locations used in the study derive from unlimed lakes and natural springs and represent original Mesolithic $^{87}Sr/^{86}Sr$ ratios in the area. The four northernmost water sample locations are selected to investigate $^{87}Sr/^{86}Sr$ ratio of the water catchment route to Lake Vesan and the southernmost and westernmost water sample locations to investigate the aquatic $^{87}Sr/^{86}Sr$ baseline in the sedimentary bedrock landscape around Norje Sunnansund. They are, from north to south, water source: VI: Lake Bredagylet $^{87}Sr/^{86}Sr$ ratio at 0.719016, V: Lake Stora Svartsjön $^{87}Sr/^{86}Sr$ ratio at 0.720029, III: Lake Vitavatten $^{87}Sr/^{86}Sr$ ratio at 0.721113, I: Hålabäck $^{87}Sr/^{86}Sr$ ratio at 0.720611, II: Barnakälla $^{87}Sr/^{86}Sr$ ratio at 0.712687 and IV: Skönabäckskällan $^{87}Sr/^{86}Sr$ ratio at 0.713100. The Vesan water catchment route is based on data from the pond and lake register, Swedish Metrological and Hydrological Institute and the lake/river liming data is based on information from the Swedish National Lime database map.

$^{87}Sr/^{86}Sr$ ratio would have differed significantly from the terrestrial signal, which is illustrated through the $^{87}Sr/^{86}Sr$ data from the water analyses of the tributaries (Fig 6). To establish the aquatic $^{87}Sr/^{86}Sr$ baseline of ancient Lake Vesan, seven archaeological water vole teeth were sampled with three measurements on each tooth (S26-S32 Figs in S1 File), apart from water vole 350 where one of the ablations struck dentine and was excluded (see S38 Table & S30 Fig in S1 File). The data obtained from these analyses (S33-S39 Tables in S1 File) provide a local "pre-contamination" aquatic bioavailable signal. However, in contrast to $^{87}Sr/^{86}Sr$ ratios obtained directly from a water source or species deriving their full diet from a particular water system, water voles are a semi-aquatic species who spend their lives on a river or lake bank (most often within a two meters range from the water's edge [101]), with a diet from both terrestrial and aquatic environments [102]. The contribution from the different plants to the diet varies somewhat across the year, but is, in general, dominated by rushes (Juncaceae) and sedges (Cyperaceae) with the most common species represented by common rush (*Juncus effusus*) followed by bulrush (*Typha latifolia*), *greater pond sedge (Carex riparia) and* great willowherb (*Epilobium hirsutum*) [103]. These are all wetland species that most often grow in direct contact with the water in wetlands or on the banks of lakes and rivers [104]. Furthermore, terrestrial grasses (Poaceae), have recently, in studies of water vole habitat use from central Wales in the United Kingdom, been shown to be of limited dietary importance (representing less than 5% of the diet in controlled feeding stations) [103]. This suggests a close relationship between the $^{87}Sr/^{86}Sr$ ratio in the water

vole teeth and the bioavailable $^{87}Sr/^{86}Sr$ ratio of ancient Lake Vesan itself. While minor discrepancies in the $^{87}Sr/^{86}Sr$ ratio within a single tooth are conceivable, i.e. corresponding to tooth development and seasonal variations in diet, the main input to the $^{87}Sr/^{86}Sr$ ratio in the water vole teeth derived from the water within ancient Lake Vesan. However, due to the inclusion of terrestrial plants in the water vole diet, the $^{87}Sr/^{86}Sr$ ratio of ancient Lake Vesan was likely somewhat more elevated than the signals obtained from the water vole teeth. Consequently, because the exact Sr contribution from aquatic or terrestrial plants, during the development of the teeth, cannot be obtained; the Sr ranges within ancient Lake Vesan cannot be confirmed solely on water vole teeth. Thereby, estimations deriving from both the sampled water sources in the Vesan water catchment route (Fig 6) and the data from the water vole teeth, provide the best available means to establish the original aquatic baseline for ancient Lake Vesan (Fig 7). The combination of un-uniform data sources and data points that are not normally distributed suggests that the traditional way of illustrating local baseline ranges, e.g. through mean values with added standard deviation [77, 105], does not adequately describe the Sr ranges of ancient Lake Vesan. Instead, robust statistics is applied using boxplots with extended variability to cover the most likely $^{87}Sr/^{86}Sr$ range of the former lake. To enhance comparability with both the singular water source measurements along the tributaries and with the previously established terrestrial baseline, obtained through bulk analysis of field vole teeth, the median value from each water vole tooth (and not each single ablation) have been used to create the boxplot from which the

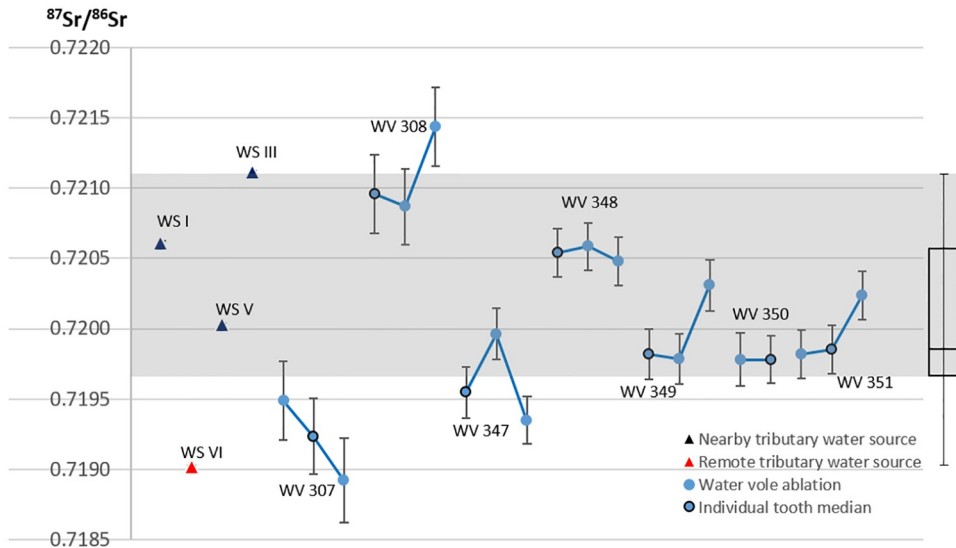

**Fig 7. Aquatic baseline estimation for ancient Lake Vesan.** Baseline (grey rectangular area) estimated from the interquartile range (25–75 percentile of the data set, using the median value for each tooth), with an added upper variability (the upper whisker representing 1.5 times the interquartile range or, as in this case, the max/min values if they are within this range). Lower variability excluded from baseline estimation. The exclusion of the lower variability is based on the dietary input from terrestrial plants in the water vole diet, which have lowered the water voles $^{87}Sr/^{86}Sr$ ratio in comparison to ancient Lake Vesan (terrestrial baseline estimated to 0.7172 ±0.008 (1SD). The exclusion of the lower variability in the aquatic baseline estimation is further motivated by the water sampling sources indicating increasing Sr ratios along the tributary system towards the former Lake Vesan (see Fig 6). Estimations based on water vole (WV) data and water samples (WS) collected along the tributary water system that once delivered water to the former lake. Dark blue triangles represent nearby tributary water sampling sources where WS I = Hålabäck, WS III = Vitavatten, WS V = Stora Svartsjön and red triangle represent a remote tributary water sampling source where WS VI = Bredagylet (see Fig 6 for location and S40 Table in S1 File for specific data). Light blue dots represent laser ablations on the water vole teeth with the blue line connecting the ablations from each tooth (data in S33-S39 Tables and S26-S32 Figs in S1 File) and black circle around a blue dot marks the median value of the combined ablations on each tooth.

aquatic baseline is estimated (Fig 7). Thereby, to estimate the bioavailable baseline at ancient Lake Vesan, the interquartile range of the data set, with added upper variability, was used (Fig 7 right). The lower variability within the data set is excluded from the aquatic bioavailable estimations. This is based on the feeding habits of water voles and their mixed diet, as it includes plants from terrestrial sources (with a significantly lower $^{87}Sr/^{86}Sr$ range of $0.7172 \pm 0.008$, 1SD), which would have lowered the $^{87}Sr/^{86}Sr$ ratio in the water vole teeth compared to if their diet would have derived solely from ancient Lake Vesan affected plants. The decision to exclude the lower variability from the ancient Lake Vesan aquatic baseline estimations is further strengthened when comparing the data from the sampled water sources along the tributary system, as their $^{87}Sr/^{86}Sr$ ratio is most elevated in the lakes closest to the Norje Sunnansund area and lowest in the lake furthest away. The interquartile range extended with the upper variability of the data set and excluding the lower variability is thereby considered the best approach to derive at the most accurate estimate of ancient Lake Vesan bioavailable baseline.

## Fish teeth data

A total of 280 individual ablations were taken from the 24 fish teeth. The majority of the samples were from tooth enamel. However, because of rapidly thinning enamel thickness distally from the tip of the fish teeth, 75 cyprinid samples included both enamel and dentine data where the enamel was too thin even for the shallow ablation lines. Similarly, 40 of the pike ablations went through the enamel and struck the underlying dentine, because of the distally thinning enamel cover (cf. S2-25 Figs in S1 File). Because of uncertainties regarding Sr isotope analyses of dentine (see the Methods), the interpretations only include the enamel-only data (Fig 8).

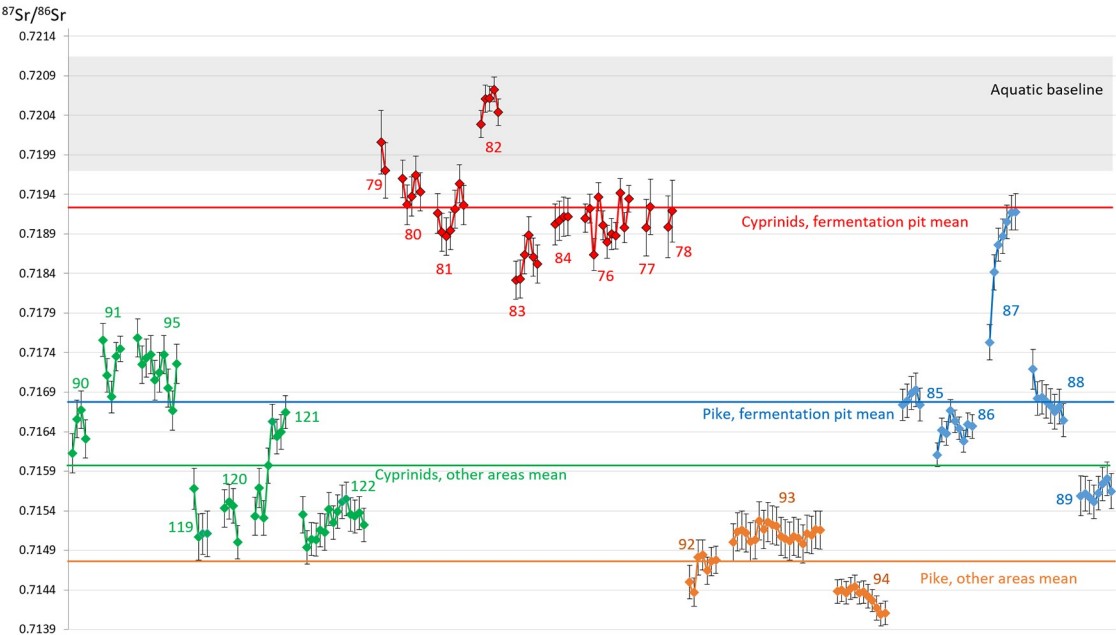

**Fig 8. $^{87}Sr/^{86}Sr$ ratios from all the fish laser ablations.** The fish from different areas and species are coloured differently for illustrative purposes; green represents cyprinids recovered from other areas (roach: 90, 91, 95, 119, 120, 121, 122), red represents cyprinids recovered from within the fermentation pit (roach: 79, 80, 81, 82, 83, 84; bream: 76; rudd; 77; dace: 78), orange represent pikes from other areas (pike: 92, 93, 94) and blue represent pikes from within the fermentation pit (pike: 85, 86, 87, 88, 89). Mean value of all measurement within each category is shown with a coloured line. The bioavailable aquatic baseline is illustrated as a grey rectangle. For a different presentation of the fish teeth data, see also S35, 36 Fig in S1 File.

As can be seen in Fig 8, there is a clear separation between the cyprinids found within the fermentation pit and those found outside of the pit. Similarly, pike found within the pit was separated from pike found outside, indicating different origins for the different populations.

If relating the cyprinid enamel data to the local aquatic baseline it shows that the different fish populations, represented by the cyprinids within and outside of the fermentation pit, are of different origin and mainly non-local. It also shows that two of the cyprinids from within the fermentation pit are local, while the others are not.

A similar, though less clear, pattern can be observed among pike. In general, the pike teeth show limited mobility during their enamel formation and they appear to have formed while under the influence of homogeneous $^{87}Sr/^{86}Sr$ ratios. The one exception to this pattern was pike 87, which displayed increasing $^{87}Sr/^{86}Sr$ values throughout its series of measurements. Similar to the cyprinids, the pikes recovered from areas outside the fermentation pit seem to have originated from other areas than the ones recovered inside the pit. The differences are not as pronounced as among the cyprinids but the pikes found within the fermentation pit have $^{87}Sr/^{86}Sr$ ratios more consistent with the cyprinids from other areas, whereas the pikes recovered from other areas have lower values. None of the pikes shows Sr ratios consistent with the local aquatic baseline.

As can be seen in Fig 8, there were no within-family overlaps between the enamel data from within the fermentation pit compared to outside it. There was, however, an overlap between the cyprinids outside the fermentation pit and the pike within the pit. Furthermore, none of the main fish groups shows any significant overlap with either the main $^{87}Sr/^{86}Sr$ range from the tributaries or the water voles, which show similar Sr ranges (Fig 9).

Regarding the mobility of the fish during the enamel development of the ablated tooth, there appeared to be some differences. The pike $^{87}Sr/^{86}Sr$ values were generally homogeneous within each tooth (apart from pike 87, which diverged from all other pikes). Similarly, the cyprinids showed limited mobility during the formation of the enamel. However, because many of the ablations struck dentine, a more limited number of enamel-only ablations was available, and thus the potential mobility signal from the enamel formation was more limited. Nevertheless, cyprinids, both inside and outside the fermentation pit, showed more elevated mobility signals than the pike during enamel formation. This is illustrated in Fig 10, where the mean value of the enamel measurements from each tooth is shown with the standard error of the mean, which accounts for both the standard deviation and the number of ablations per tooth. The error bars are shorter for pike compared with the cyprinids, again apart from pike 87, which displayed a completely different and elevated mobility pattern.

For pikes, only pike 87 was the exception to the general trend of low pike mobility. The cyprinids also showed rather homogeneous enamel Sr data signals, albeit with somewhat larger discrepancies than pike, indicating that, within the confines of their particular environment, they were in general more mobile during enamel formation than the more sedentary pike. The cyprinids from outside the fermentation pit displayed slightly larger error bars than those recovered from within the pit. From the fermentation pit, cyprinids 77 and 79 showed somewhat larger error bars, although for both these fish only two enamel-only ablations could be made on each tooth, which increased the standard error of the mean. For the cyprinids outside the fermentation pit, cyprinid 121, in particular, stands out as having been more mobile during enamel formation, with comparably large variations in Sr values.

## Discussion

A pit from Norje Sunnansund provides the world's earliest known evidence of the practice of fish fermentation by humans [62], and for this reason alone the site is interesting. However,

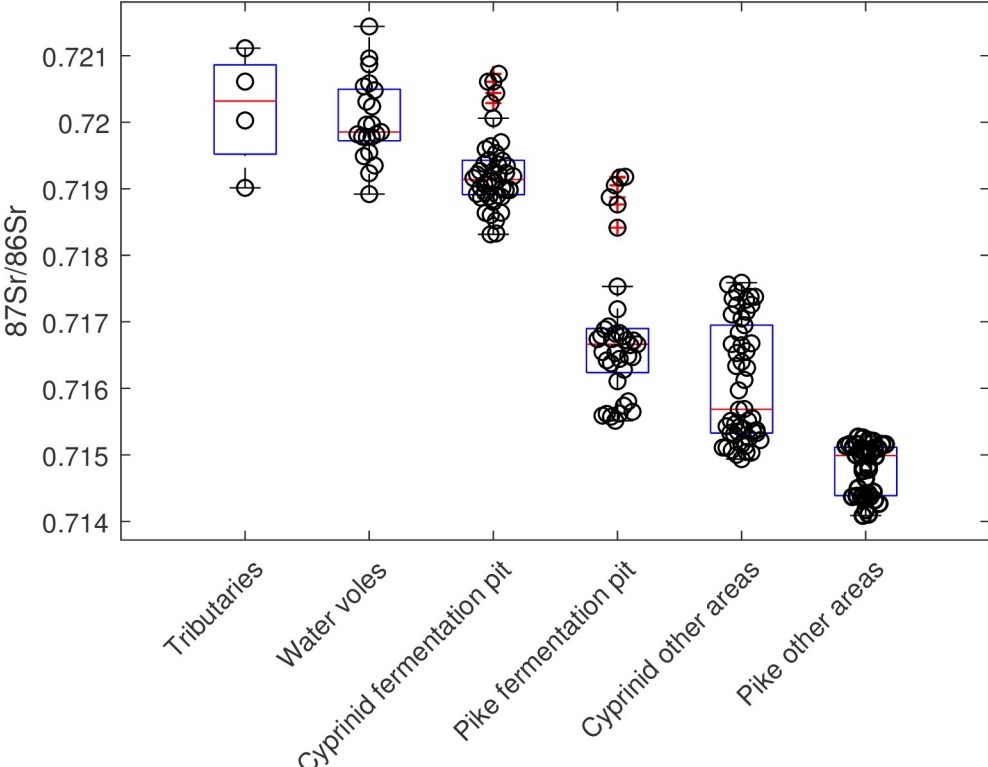

**Fig 9. Boxplots of the $^{87}$Sr/$^{86}$Sr data obtained from the tributaries along the water catchment route, the water voles and the four fish categories.** The median of each category is shown as the internal horizontal line within each box bounded by the upper and lower quartiles with whiskers representing the variation at 1.5 times the interquartile range (or the max/min values when they are within this range). Each data point is added as a circle and any outliers are highlighted with a + mark.

our understanding of Mesolithic Scandinavian hunter-fisher-gatherers (and indeed ancient foragers around the globe) can be expanded by many other insights gained from Norje Sunnansund, such as the unique organic remains that indicate year-round occupation (with an increased presence during winter) [29], and the large quantities of recovered fish bones that indicate that advanced fisheries were established in Scandinavia in Early Holocene [19]. Evidence of other activities has been discussed elsewhere, such as targeted hunting strategies [29] and territorial displays through possible excarnation practices [19]. The site is also important because it appears to have been influenced by both eastern and western stone and bone technological traditions [64, 106, 107], suggesting wide-spread social networks. Sr bulk data from 12 human teeth, representing a minimum of 10 individuals, indicate a generally local residency, but with some individuals probably coming from more remote northern areas [92].

The diverse evidence from Norje Sunnansund includes indications of sedentism and year-round occupation, with delayed-return subsistence practices made possible through fish storage. Furthermore, contemporaneous sites from southern Scandinavia are located inland. No other Scandinavian coastal site with such organic preservation has been found or excavated, because the transgression following the last Ice Age has left coastal regions submerged [108–110] and therefore difficult to find and recover on a similar scale as at Norje Sunnansund. The evidence gained from Norje Sunnansund is therefore hugely important for our understanding of Early Holocene human foraging societies.

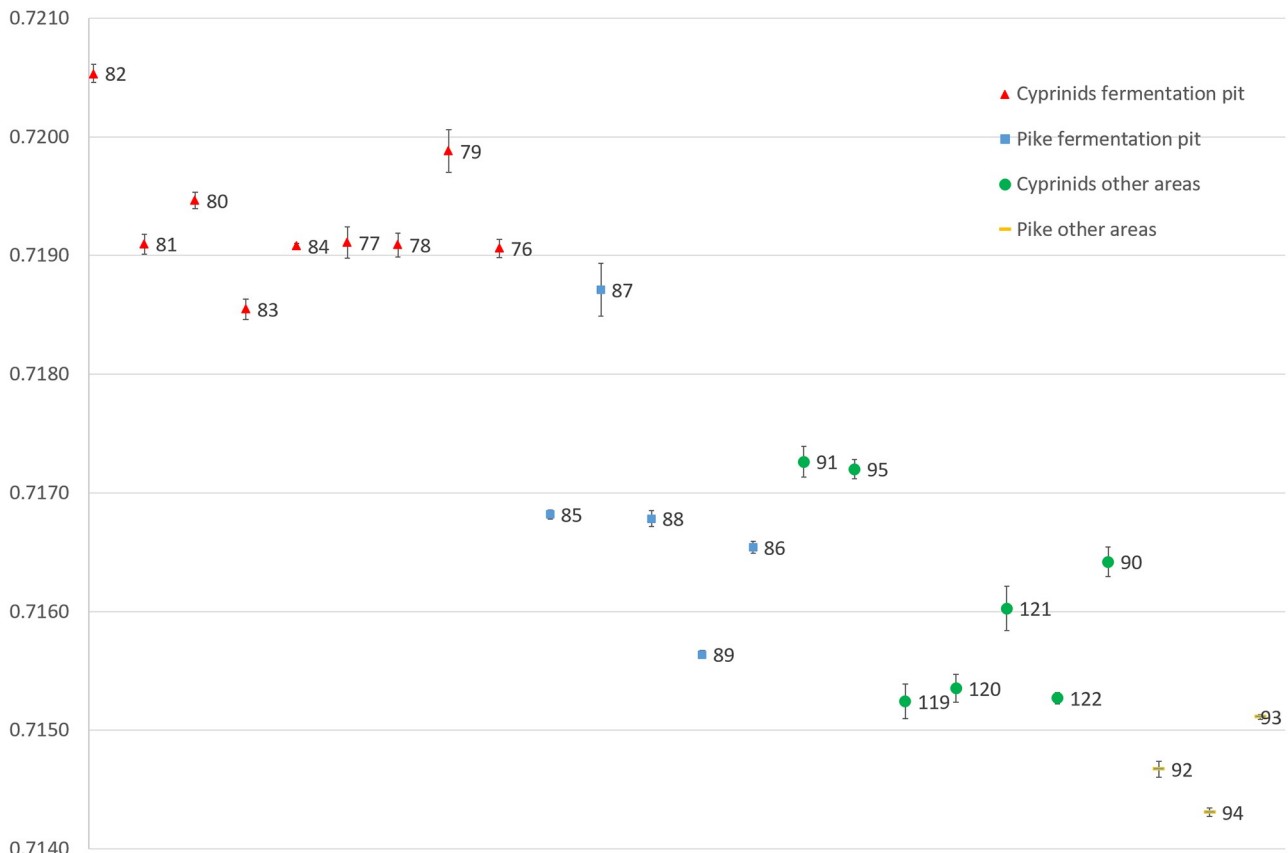

**Fig 10. Mobility signals for the four fish categories.** Illustrated as the mean and standard error of the mean, for the combined enamel ablations from each tooth.

The evidence provided by the Sr isotope analyses furthers that understanding and provides a much-needed insight into how human Mesolithic societies utilized fish mobility patterns and natural behaviour. The estimation of the local aquatic baseline proved challenging both because Lake Vesan has been drained and no longer exists and because both water sources and soil in the area have been subjected to extensive liming. By locating the few available water sources along the tributary water catchment route leading to the former Lake Vesan and by sampling archaeological water voles and relate the Sr ratios obtained from their teeth to their diet, it was possible to estimate the original Sr baseline within the former lake (Fig 7). This showed that two of the cyprinids were local to the former lake while the other fish came from other areas (Fig 8) and did not show an overlap with the data obtained from the water sources or the water voles (Fig 9).

Other distinct patterns were revealed by the LA-MC-ICP-MS data, with different regions of origin indicated for the four fish categories analysed. The first regions of origin were indicated by the Sr data for the cyprinids recovered within the fermentation pit. It showed that two fish were local (roach 79 & 82), while the remaining seven cyprinids in this category originated from areas of somewhat lower Sr ratios than what was locally available in ancient Lake Vesan. The cyprinids found outside the fermentation pit and the pikes found within the pit display overlapping $^{87}Sr/^{86}Sr$ ratios consistent with an area with a lower Sr ratio than the cyprinids found within the fermentation pit or that of the local aquatic baseline. The last region of origin

is indicated by the pikes found outside the fermentation pit. Though there is overlap between some of the cyprinids from other areas, these pikes have generally lower $^{87}Sr/^{86}Sr$ ratios than the other fish categories studied.

The possible identification of three separate regions of origin (four if including the two local cyprinids) is intriguing, considering the species diversity of the areas across the site from which the fish were recovered. If related to the bioavailable Sr ratios of southeastern Sweden, the $^{87}Sr/^{86}Sr$ ratios of the cyprinids from the fermentation pit suggest that the majority of them came from coastal areas of the Baltic Sea to the northeast of Norje Sunnansund, between the Blekinge archipelago and the Kalmar strait area (Fig 11). However, although only represented with one water sample with lower Sr ratios than what the surrounding bioavailable data and bedrock suggest (and thus possibly affected by liming even though the sampled lake is not listed as having been limed), it is also possible that they originated far up at the beginning of the tributary system (see Sr ratios in Lake Bredagylet Fig 6).

The cyprinids from the fermentation pit mainly consist of roach; which, today, has a distinct potamodromous migration pattern. Roach have their major growth period during the warmer months of the year, when zooplankton, zoobenthos, detritus, epiphytes, phytoplankton and macrophyteszooplankton, which form the main part of their omnivorous diet [111], are abundant [112]. It is also during this period (spring to autumn) that their teeth (and other skeletal material) are affected by the environment. Because little growth takes place during the

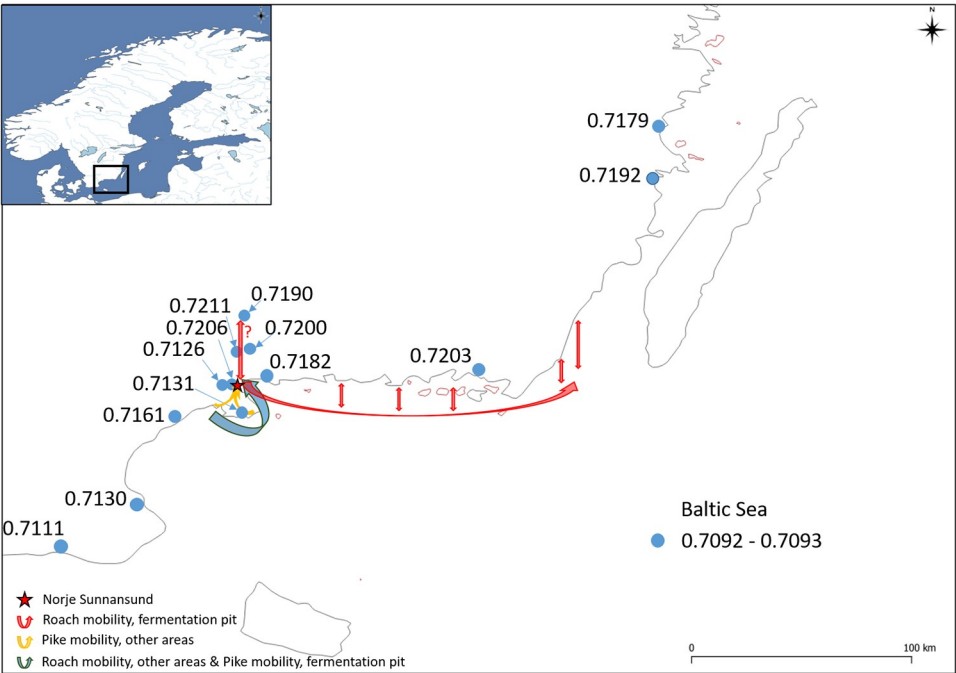

**Fig 11. Suggested centres of origin and generalized mobility patterns for pike and roach from Norje Sunnansund, based on their $^{87}Sr/^{86}Sr$ ratios.** Baseline data derived from water samples according to [72] and this study, and are illustrated with their unique values and a blue dot marking the sampling location. Sr ratios from the cyprinids recovered in the fermentation pit, suggests a migration from coastal areas of the Baltic Sea to the east and north of the site and into the stream and lake at Norje Sunnanund during autumn. The Sr ratios obtained from the cyprinids recovered at other areas of the site suggest late spring spawning mobility from Baltic Sea coastal areas to the south of Norje Sunnansund. The Sr ratios obtained from the pikes recovered at other areas of the site suggest early spring spawning migration from the close by rivers and estuaries to the south of Norje Sunnansund. Lastly, the Sr ratios obtained from the pikes recovered within the fermentation pit suggest mobility from areas along the Baltic Sea coast during the autumn to ancient Lake Vesan where they were caught as by-catch to the mainly targeted migrating roach.

winter months [113], when there is a decline in available food, this implies that the main tooth formation in roach teeth takes place during the summer. Consequently, the obtained Sr signals from each roach tooth most likely represent the roach's summer location. Furthermore, roach congregate in shallow streams during spring for spawning, after which they migrate to their summer location and then, in the autumn, migrate to sheltered areas, wetlands and smaller streams, where they are 'safe' from predation during winter [113]. Consequently, the non-local Sr signals obtained from almost all of the fermented cyprinids indicate that they were caught during one of these congregations, suggesting that the humans were primarily using migrations and spawning activities to catch huge quantities of fish that were then fermented and stored in bulk. Furthermore, the presence of two local roaches within the fermentation pit suggests that the fishing activities were conducted within ancient Lake Vesan; which support the interpretation of the area as a fish resource extraction hot spot [19].

Although from a different region of origin than the cyprinids from within the fermentation pit, the cyprinids from other areas of the site also had non-local $^{87}$Sr/$^{86}$Sr ratios. If related to the bioavailable Sr in the area of southeastern Sweden (Figs 5 & 6) it suggests a centre of origin to the south of ancient Lake Vesan, along the Baltic coast (Fig 11). If combined with roach spawning activities in shallow streams it further highlights that different cyprinid migrations were targeted at Norje Sunnansund and that the roach recovered outside the fermentation pit were caught in the stream(s) leading to ancient Lake Vesan during their spawning in late spring. The pike within the fermentation pit had Sr ratios similar to those of the cyprinids from other areas of the site, which, consequently, suggest a non-local origin with Sr ratios matching the baseline along the Baltic coast to the south of the Norje Sunnansund area.

An additional region of origin is hinted at for the pike from other areas of the site, with the lowest Sr ratios among the different fish categories in the study. If relating their Sr data to bioavailable Sr in the area, the two water sources sampled in the areas of sedimentary rocks to the east and south of Norje Sunnansund (Fig 6) have lower Sr ratios than the pikes from other areas. However, the samples both derive from natural springs and were not part of the water system supplying the former lakes and streams with water. Given that ancient Lake Vesan had a much higher Sr ratio than the local terrestrial baseline, due to its water supply deriving from tributaries of older bedrock with elevated Sr ratios (Fig 6), it is plausible that the original streams and lakes in these sedimentary rock areas had somewhat more elevated Sr ratios than what was obtained here. Consequently, and because the Sr ratios obtained for the pikes from other areas are only somewhat more elevated than the ratios obtained from the sampled natural springs, these pikes likely originated from the nearby sedimentary rock areas to the south of ancient Lake Vesan (Listerlandet). Apart from this area, the closest location with a Sr baseline matching the data obtained from the pikes outside of the fermentation pit corresponds with southeastern Scania, which is almost 100 km to the south, if following the coastline (cf. Fig 11), which makes an origin in the water systems of Listerlandet more plausible.

Pike and roach are both spring-spawning species. Pike normally spawn in early spring closely following the thawing of the ice (in southern Sweden today often in March) and roach spawn somewhat later when the water temperature has increased (often during a short period between April and June) [114]. Because the Sr signals diverge between different depositions at the site, it provides a strong argument that different periods/seasons of catch are represented. While this pattern can probably be translated into the season of catch for the pike from outside of the pit (early spring) it is not as straightforward for roach. Roach, particularly if living in brackish water, does congregate in freshwater systems to spawn during late spring but is also known to migrate during autumn [115], which, at specific locations, creates large roach congregations [116]. The different $^{87}$Sr/$^{86}$Sr values for the cyprinids within compared with outside the fermentation pit indicates different cyprinid populations of which all but two roach were

non-local to ancient Lake Vesan. This suggests that roach migrations were targeted both during late spring spawning and during autumn migrations.

Several inferences regarding the different roach population season of catch, and the subsistence strategies behind the choices, can be made. First, the fermentation process, used to enable long term storage at the site, was done without the use of salt. This is evident as salt would not have been available to Early Holocene southern Scandinavian foragers living in a freshwater environment (the earliest evidence of salt use in Europe is from south-European Neolithic groups in the 8th millennium BP [117], while the earliest known evidence of salt-use in the world is from north African foraging societies around 9000 cal BP in [118]). Without using salt, the fermentation process would have required cold temperatures to avoid the development of harmful bacteria [62]. The utilization of the roach autumn congregation (rather than the roach spawning period in the spring) is therefore supported for the fish within the fermentation pit, as safe fermentation would have been facilitated by carrying out the process during the coldest part (late autumn and winter) of the year. Furthermore, since the fermented fish was likely caught during a limited time, though likely at a yearly reoccurring basis [62], the fish bones from within the fermentation pit can be seen as a temporally limited event.

The fish bones from the other areas of the site likely represent different catches, from different congregation events at different times of the year, suggesting optimal use of species-specific seasonal abundance. Thus, following known ecological prerequisites for the two investigated species [114], the pike from other areas represent pikes caught in early spring while the roach in the other area represents roach spawning in late spring. Consequently, the patterns we see in the fish Sr data show that the Early Holocene foragers from Norje Sunnansund were able to make full use of seasonal fish congregations, during different periods of species-specific agglomeration at different times of the year, to generate a surplus.

## Concluding remarks

The observed mobility patterns from the different fish categories in the study indicate that the fermented roach was probably caught during its migratory congregation in the autumn, while the more limited amount of pike recovered in the pit was likely a by-catch to the mainly targeted roach. Furthermore, the Sr data suggest that both the pike and the cyprinids from the other areas where also of non-local origin, which suggest that different fish congregations where exploited on different occasions during different times of the year. Particularly regarding the fermented fish, which were caught to create a large scale and long-term storage, this has further implications.

Recurring and regular access to food is a prerequisite for human communities to develop a sedentary lifestyle. This can be achieved through mass exploitation and storage. If the community is unable to store sufficient supplies during times of surplus, more mobility-orientated foraging strategies are required. The predictable nature of migrating salmonids, and the human ability to 'cultivate' and optimally exploit it [119], has often played a pivotal role in discussions of why the foraging societies from the northwest coast of North America developed into complex hierarchical communities [6, 10, 13, 17], with population increases beyond what would be achievable in non-aquatic-dependent foraging societies at northern latitudes [9, 120]. However, the vast majority of the fish from Norje Sunnansund are freshwater species (roach in particular) and less than a per mille of the identified fish bones belong to migrating anadromous species [29]. The indications of a year-round occupation, combined with the almost complete absence of migrating salmonid bones at Norje Sunnansund, suggests that roach may have played a role similar to that of salmon among North American societies. Consequently, and given that roach represents about 80% of the fish bones found in the fermentation pit, roach

need not only meet the criterion of availability during congregation but also that of predictability, if they are to be relied on (through storage) by a sedentary community during the leaner months of the year. Interestingly, and in support of these requirements, roach is in many ways similar to traditionally used migrating fish as they are also known to return to their birthplace. Studies from Norwegian lakes have shown that more than 90% of spawning roach return to the same spawning locations the following year [121]. In addition, the use of seasonal congregations among aquatic species other than salmonids is well known and also evident in the archaeological record from the northwest coast of North America [122], emphasizing that the specific ecological setting needs to be considered when discussing the prerequisites for human utilization of temporally limited but abundantly reoccurring aquatic resources.

Here it has been demonstrated that it is possible to detect the origins of individual fish, and in so doing it is also possible to show that Scandinavian Mesolithic foragers were able to utilize seasonal potamodromous fish migrations. Consequently, these foragers were able to practise delayed-return subsistence strategies and generate a surplus stock that could be used in times of scarcity, reducing residential mobility. The capacity to exploit seasonal fish congregations and the ability to store large amounts of fish throughout the winter through fermentation is a clear signal that, at least given favourable circumstances, Early Holocene societies were becoming more sedentary. With increasing sedentism, if supported by predictable and sustainable amounts of food, human foraging societies tend to experience increasing populations [9]. When these societies are located in temperate zones where resources are abundant only in particular areas and during limited periods of the year, claims of resource ownership often increase along with a territorialization of the landscape [11]. Indeed, the territorialization of the Early Holocene landscape has already been suggested for Mesolithic Scandinavian communities (cf. e.g. [19, 41, 47] and is supported by evidence of large populations among the Late Mesolithic south Scandinavian foraging communities of the Ertebølle culture [61].

While only one fermentation pit has been identified at Norje Sunnansund, there are indications that it was probably just one of many similar facilities used to conserve and store large catches of fish [29]. There is also evidence that the same fermentation pit was re-used many times before finally being abandoned [62]. It stands to reason that if a community can utilize the full potential of seasonal fish migrations and successfully store the food for extended periods, this is not a one-time culinary experiment or something that is only carried out a few times and then abandoned. It is a subsistence strategy that will endure and is indeed likely to expand, especially given the wide contact network indicated by the lithic material, the bone tool technology and the human Sr isotope signals [63, 64, 92, 106, 107]. It is therefore plausible that what can be observed at Norje Sunnansund took place elsewhere as well.

The insight that people were able to use fish spawning and migration in freshwater environments is important, and, while it does not negate the importance of the eutrophic and productive freshwater lakes themselves, cf. e.g. discussions in [19, 35, 123], it places them in a seasonal context within a system of logistic mobility patterns, cf. e.g. [11, 124, 125]. Given the observations presented here, it is worth considering that Early Holocene human societies along the coastline might have been relying on fish migrations/congregations on a much larger scale than has hitherto been considered. Furthermore, different means to facilitate long-term storage were, in this case, probably used more often and in larger quantities than we have so far found evidence for in the archaeological remains. While other means of storage has not been verified at Norje Sunnansund, considering that the inhabitants exploited seasonal fish congregations of different species throughout the year, fermentation was probably only one of many alternatives used to conserve food and store it long term (cf. e.g. discussions in [126–134]).

As shown here, the ability to trace migration in non-diadromous fish species and connect it with human exploitation is promising, and it should be possible to trace similar types of fish

exploitation elsewhere. At Norje Sunnansund this exploitation is connected with evidence of large-scale storage. As storage methods are typically difficult to detect in archaeological remains, Sr mobility signals, as observed in the different areas of Norje Sunnansund, could be used as a model to detect similar utilization of fish spawning. Consequently, the methodology presented here could facilitate the detection of past foraging societies' ability to benefit from natural occurring fish congregations. If this is also connected with mass catching devices or substantial amounts of fish bones, it might be an indication of active resource management and a striving for affluence, both at Norje Sunnansund and further afield. This new methodology can therefore be used to investigate whether this is indeed the case among Scandinavian Mesolithic foragers in general; while, at the same time, it may enable similar discussions elsewhere.

## Supporting information

**S1 File.**
(PDF)

## Acknowledgments

We would like to thank the three reviewers for their insightful comments. We acknowledge and thank the curators at Blekinge Museum for permission to analyse and sample bone remains in their charge and for providing and permitting the use of photographs and images concerning Norje Sunnansund. This is Vegacenter publication number #035.

## Author Contributions

**Conceptualization:** Adam Boethius.

**Data curation:** Adam Boethius, Melanie Kielman-Schmitt.

**Formal analysis:** Adam Boethius.

**Funding acquisition:** Adam Boethius.

**Investigation:** Adam Boethius.

**Methodology:** Adam Boethius.

**Project administration:** Adam Boethius.

**Resources:** Adam Boethius, Mathilda Kjällquist, Melanie Kielman-Schmitt.

**Writing – original draft:** Adam Boethius.

**Writing – review & editing:** Adam Boethius, Mathilda Kjällquist, Melanie Kielman-Schmitt, Torbjörn Ahlström, Lars Larsson.

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
