## [Decision Letter · Decision Letter 0]

21 Jul 2020

PONE-D-20-15007

Early Holocene Scandinavian foragers on a journey to affluence: Mesolithic fish exploitation, seasonal abundance and storage investigated through strontium isotope ratios by laser ablation (LA‐MC-ICP‐MS)

PLOS ONE

Dear Dr. Boethius,

Thank you for submitting your manuscript to PLOS ONE. After careful consideration, we feel that it has merit but does not fully meet PLOS ONE’s publication criteria as it currently stands. Therefore, we invite you to submit a revised version of the manuscript that addresses the points raised during the review process.

The paper presents a very original study that pushes forward the perspectives on Sr isotopes analysis on archaeological fish. However, as for every challenging material and method, the authors need to thoroughly consider some important aspects to strengthen their study. First, the (unfortunately) very likely diagenetic contamination of dentine means that the data have to be examined apart from the enamel and I encourage the authors to follow the recommendations of reviewer 1 and 2 about it. Reviewer 2 and 3 point to technical aspects such as the Sr voltages that need clarification. Moreover, a couple of information and possibly new measurements are recommended to robustly rule out any Ca/Ar PO interferences. The constructive suggestions from the reviewers will help to achieve a secured presentation of the data, which will raise high interest in the audience. 

We look forward to receiving your revised manuscript.

Kind regards,

Dorothée Drucker

Academic Editor

PLOS ONE

Journal Requirements:

2. In your manuscript, please provide additional information regarding the specimens used in your study. Ensure that you have reported specimen numbers and complete repository information, including museum name and geographic location.

For more information on PLOS ONE's requirements for paleontology and archaeology research, see https://journals.plos.org/plosone/s/submission-guidelines#loc-paleontology-and-archaeology-research.

3. Thank you for including the following funding statement on your acknowledgements section; "We are grateful for the financial support from the Swedish Research Council (VR-2019-02975) and Birgit och Gad Rausings Stiftelse för Humanistisk forskning. "

Reviewers' comments:

Reviewer's Responses to Questions

**Comments to the Author**

1. Is the manuscript technically sound, and do the data support the conclusions?

Reviewer #1: Partly

Reviewer #2: No

Reviewer #3: No

2. Has the statistical analysis been performed appropriately and rigorously? 

Reviewer #1: Yes

Reviewer #2: N/A

Reviewer #3: No

3. Have the authors made all data underlying the findings in their manuscript fully available?

Reviewer #1: Yes

Reviewer #2: No

Reviewer #3: No

4. Is the manuscript presented in an intelligible fashion and written in standard English?

Reviewer #1: Yes

Reviewer #2: Yes

Reviewer #3: Yes

5. Review Comments to the Author

Reviewer #1: The manuscript titled “Early Holocene Scandinavian foragers on a journey to affluence: Mesolithic fish

exploitation, seasonal abundance and storage investigated through strontium isotope

ratios by laser ablation (LA‐MC-ICP‐MS)” by Boethius et al. describes the use of strontium isotope ratios to identify origin and migration patterns of fish from the archaeological site of Norje Sunnansund, southern Sweden. Overall, 16 cyprinid and 8 pike teeth, from within and from outside of fermentation pits were analyzed using laser-ablation techniques and the results interpreted to support the idea of an year-round fishery with intensified use and high catches during seasonal fish migrations.

Overall, the article is well written and well organized with clear sections outlining the research questions and providing the necessary background and methods. The analytical and statistical methods are sound, and the author have met all ethical standards during data collection.

However, I have some major concerns about the discussion and interpretation of the data that I would like to see addressed.

1) The authors need to decide if the trust the dentine “contaminated” values or not. In several parts of the manuscript the authors rightfully point to the many potential problems in interpreting the dentine data (interferences, Rb content, timing of formation, potential diagenesis,…) and based on this I would suggest to remove these data from the main manuscript and discussion and add them to the supplement. In addition, the uncertain proportion of enamel-dentine mixing make these data incredibly difficult to interpret if the timing of formation of these materials is different. It appears to me that the discussion is still very useful without the dentine data and would be more streamlined and in tune with the archaeology focus. This would change figures 6-11 which as currently formatted are not easy to interpret and change the text of the entire results section. This could focus on potential migration paths and origins of the fish using the known baseline data for Sweden and the Baltic Sea.

2) Local baseline. I don’t think the vole teeth are a good local baseline for fish teeth from a nearby lake and river? I take it the lake has been drained and the river is not accessible anymore? Since a lot of the paper discussion hinges on local vs seasonal migration you will need to establish local values more robustly. Maybe based on the geology of the proposed watershed? Based on known local fish bones? It’s tricky but how would vole teeth be appropriate for a watershed and with only three samples with all different values who is to say the local range is not much larger? Also mean +-sd from 3 samples is not very meaningful. All this needs to be better established and supported by baseline data for the discussion to be more robust.

3) If the fish migrated as suggested in figure 12 than why do you not see any Baltic Sea values in the enamel samples?

Minor comments

4) Section - Prerequisites for studying provenance and mobility using strontium isotope ratios. Since this paper combines ecological and archaeological interpretations I would expect a short introduction to Sr isotopes in fishery research here so citing some studies using otoliths. How are isoscapes for these types of studies normally constructed and what kind of information can be gained? Sean Brennan’s work on building robust water isoscapes for pacific salmon could be cited here in addition to more regional papers focusing the Baltic (e.g., Glykou et al., 2018).

5) How did you ensure that each tooth (from the same fish species) represents a unique fish?

6) Section - Fish tooth development. Could you expand on your best assessment here what time frame these teeth capture? Is it Months? Years? Are there any morphometric features that can provide insights into this?

Just a comment here that the supplementary materials are excellent! This should be required more often and its great to see the authors provide all these details!

Line 26-29: Not clear to me what “the mobility signal of the fish is specific and follows general patterns” means.

Line 89: Maybe provide more examples instead of “etc.”

Line 107: To clarify today the site is not near the river and the lake has been drained? Some more information about the site today vs before would be useful for a not familiar reader.

Line 280-285: Add the number of standards run and what the error represents to the text. (e.g. n=, +-2sd)

Figures general: I strongly suggest to redo these figures in R or python (or any other stat plotting software), especially the boxplots would be much nicer using one of those software packages. Also avoid red and green together.

Figure 1: I suggest swapping the maps around. The regional map showing the site, river, and lake as the main part of the figure and a small insert showing the location in Sweden. It would be very useful to add other major rivers and lakes to this.

Figure 5: Need to add the Baltic sea values to this maybe from Glykou et al., 2018:figure 1?

Figure 6: Sort by species and fermentation rather than by sample number?

In summary this paper represents a significant contribution to combining archaeological and ecological research and I expect it will be of great interest to a wide scientific audience once the mentioned concerns are addressed but as it currently stands I suggest it needs major revisions.

Reviewer #2: I am sorry to say that as the data is currently presented, I am not sure I trust the results and would be wary of citing the paper. As the paper is currently presented, the use of dentine data mixed with enamel data here is concerning, and unless much stronger justification can be given, I suggest that all the dentine data and mixed enamel-dentine data should be removed and the interpretations re-written based solely on the enamel-only data. My reasons for thinking this are as follows:

1. Some of the dentine-contaminated tooth profiles move towards the bioavailable range quoted for the site, but not in every case. This is potentially interesting, but only if the stated baseline bioavailable range is fully representative for the site and the burial contexts. Therefore, more information is needed on how the bioavailable 87Sr/86Sr baseline range for the site was established. Currently, a range is given (0.7166 to 0.7180) with a reference to three rodent teeth, but is this enamel/dentine/whole tooth, where specifically were the teeth found on the site, and are they archaeological or modern, burrowing rodents or not? Also, what is the site stratigraphy and could the “fermentation pit” have been dug through different sedimentary horizons/geological substrates, with different bioavailable ranges? Possibly this information is in some of the cited references but it needs to be given here, at least in summary form as identifying the local bioavailable range correctly is critical to the interpretations (see next comments).

2. Dentine is porous and typically equilibrates with the burial environment – this fact is used in many studies to compare enamel and dentine values to identify migrants. Enamel-dentine comparisons are also common when studying large teeth (cows, sheep etc.) with much thicker enamel than the fish teeth reported here, precisely because the dentine equilibrates. The argument that dentine from tiny fish teeth, with very thin enamel, can be preserved without contamination, needs much stronger justification here if the dentine results are going to be treated as giving an original uncontaminated signal.

3. Dentine-only data that falls outside the local bioavailable range would be a good indication that the dentine is preserving something meaningful. But at the moment, the data is presented only as enamel, and enamel-dentine mixed, meaning it is not possible to know if this is the case. The images of the sampled teeth suggest that there should be some dentine-only data, and this needs to be collated and analysed specifically in its own right.

4. Importantly, if the dentine is contaminated with Sr from the burial environment, data from mixed enamel-dentine samples is essentially meaningless as it is impossible to know how much the contamination has affected the results, and it should be removed entirely from the dataset.

Aside from the issues with the dentine data, I also think a comment needs to be inserted justifying the notion that the Sr ratio in the very thin enamel is itself unaffected by diagenetic contamination. In my experience it is common for the outer few 10s or 100s of microns of tooth enamel to be contaminated on every enamel surface (including at the enamel-dentine junction). This goes for teeth I have analysed by laser ablation from species with both large teeth with thick enamel and small teeth with thin enamel. Given the enamel in the fish teeth is so thin, it would theoretically be quite easy for the contamination to permeate the entire tooth, including both enamel and dentine. The spread in enamel-only values is good evidence that the enamel is uncontaminated, but some acknowledgement that this potential issue has been thought about is needed.

The Supp info gives some summary information for each line sampled in each tooth, but this is not sufficient to judge the data. Specifically, the tables give “total Sr beam” in volts for each line measured in the teeth, but I could not find an explanation of what this represents (masses 84-88 of all integrations??) and it is not useful for assessing the data because it relates mainly to the length of the line and not the data measured. Rather, please modify these tables to show number of integrations / datapoints for each line to have a clearer idea of what the SE and SD relate to (n); and also a MEAN 88Sr (not total), which might give a much better idea of when the concentration of Sr in the sample is increasing and the dentine/enamel are contaminated.

Also on identifying contamination, 89Y is commonly used in laser ablation studies to monitor for rare earth element contamination (e.g. Wilmes et al. 2016, whom you cite), and can also be a useful indicator for the presence of diagenetic strontium – if high concentrations of REEs have leached into the enamel from the burial environment, there is every chance that elemental Sr has as well. Did you measure 89Y and if yes what does this show? If not, another possibility might be exploring Sr intensities in enamel-only and dentine-only data, and comparing these groups. This is something that should be included, at least in the Supp Info but probably needs a mention in the main article too.

Figure 6 – group the samples according to where they were found (inside or outside the fermentation pit). This would make it a lot easier to read.

Figs 7 and 8 – it took me a long time to recognise what you meant by “shading”. Perhaps “thickened line” is a better description (this may be a result of how the graphs appeared on my screen).

Figure 12 needs a scale

Reviewer #3: I am unfamiliar with the archaeology of this location and period so I will restrict my comments to the quality of the LA strontium isotopic data.

Unlike conventional solution Sr isotope analysis, where ion exchange chromatography is used to remove most of the elemental/molecular interferences, LA Sr isotopic analysis suffered from a whole range of potential interferences. The authors use standard procedures to mitigate most of these (REEs Rb K etc), though I have some concerns about whether then have demonstrated that they have eliminated the Ca/Ar PO interference. This interference (predominantly on mass 87) has the potential to cause a positive offset on measured 87Sr/86Sr of several 000s ppm which would be highly significant given the c. 6000 ppm range of the authors’ data. The magnitude of the interference will vary with Sr concentration in the analyte but not with sample throughput (e.g. from varying laser spot size or repetition rate).The author’s attempt to eliminate this by careful control of oxide formation, and claim that they have been successful through the measurement of two internal standards of a shark spine and a hare’s tooth which show negligible offset from their known Sr isotopic values. This is a standard approach, but only valid if the Sr concentrations of the standards is equal to or lower than the Sr concentrations of the samples. This has not been demonstrated in this paper. The issue is further confused by the different ablation conditions used for standards and samples, rending Sr voltages difficult to relate to concentrations, and there is some confusion what the Sr Voltages quoted actually represent. Are the Vs integrated over the whole run, or averaged; it is unclear why Sr V is reported as ‘total beam average’ for the standards and ‘total beam’ for the samples? If both are meant to be the former, then the difference in spot size would mean c. 3-4x greater voltage per Sr concentration for the samples than the standards. If the latter is the V integrated over the whole run then the difference is far greater than this.

The LA results may well be unaffected by a Ca/Ar PO interference, but the authors need to robustly demonstrate this before I would recommend publication. I would suggest the authors:

- Include details of the Sr concentrations of the standards

- Remeasure the standards and two or three of the samples under the same laser and MS conditions, to calibrate and calculate a semi quantitative Sr concentration for all samples. Note, this procedure does require re-measuring of both as variations over time in instrument sensitivity precludes an extrapolation of a new measurement solely of the standards (e.g. at 150 micron spot) to previous analyses.

- Demonstrate that the neglibile LA-TIMS/seawater offset measured in the standards will also be the case for samples as the standards have similar or lower Sr concentrations.

The authors might also look at the variation in Sr isotopic values with Sr V where the voltage (and hence the Sr concentration) varies significantly. There does seem to be a downward trend in Sr isotopic values with increasing V for a couple of samples I looked at in detail (121 and 95). This might be attributed to the CaPO interference, though there are other potential explanations.

6. PLOS authors have the option to publish the peer review history of their article (what does this mean?). If published, this will include your full peer review and any attached files.

Reviewer #1: **Yes: **Malte Willmes

Reviewer #2: No

Reviewer #3: No

---

## [Author Response · Author response to Decision Letter 0]

13 Nov 2020

Dear editor and reviewers, thank you for the insightful comments. We have now addressed all comments by the reviewers and followed the formal guidelines of PLOS ONE. We have also provided information about the specimens and the permissions. Please see our colour coded replies at the end of the built pdf for our replies to the comments from the reviewers.

best regards

Adam Boethius

---

## [Decision Letter · Decision Letter 1]

17 Dec 2020

PONE-D-20-15007R1

Early Holocene Scandinavian foragers on a journey to affluence: Mesolithic fish exploitation, seasonal abundance and storage investigated through strontium isotope ratios by laser ablation (LA‐MC-ICP‐MS)

PLOS ONE

Dear Dr. Boethius,

Thank you for submitting your manuscript to PLOS ONE. After careful consideration, we feel that it has merit but does not fully meet PLOS ONE’s publication criteria as it currently stands. Therefore, we invite you to submit a revised version of the manuscript that addresses the points raised during the review process.

All the comments on the previous version have been addressed. However, reviewer 1 recommends some additional modifications that should help the authors to smooth their final version, especially by separating more clearly the results and discussion sections, combining or modifying some of the figures and streamlining the conclusion and some parts of the narrative. I encourage the authors to proceed with these last modifications and to provide an accordingly amended version of their manuscript allowing the final acceptance of the paper.

We look forward to receiving your revised manuscript.

Kind regards,

Dorothée Drucker

Academic Editor

PLOS ONE

Reviewers' comments:

Reviewer's Responses to Questions

**Comments to the Author**

1. If the authors have adequately addressed your comments raised in a previous round of review and you feel that this manuscript is now acceptable for publication, you may indicate that here to bypass the “Comments to the Author” section, enter your conflict of interest statement in the “Confidential to Editor” section, and submit your "Accept" recommendation.

Reviewer #1: (No Response)

Reviewer #2: All comments have been addressed

2. Is the manuscript technically sound, and do the data support the conclusions?

Reviewer #1: Yes

Reviewer #2: Yes

3. Has the statistical analysis been performed appropriately and rigorously? 

Reviewer #1: Yes

Reviewer #2: Yes

4. Have the authors made all data underlying the findings in their manuscript fully available?

Reviewer #1: Yes

Reviewer #2: Yes

5. Is the manuscript presented in an intelligible fashion and written in standard English?

Reviewer #1: Yes

Reviewer #2: Yes

6. Review Comments to the Author

Reviewer #1: Comments to the Author

The authors have responded to the comments by all three reviewer’s and it is clear that a lot of work has gone into these revisions. Specifically, the main problem of the dentine values seems to have been taken care off. However, I still have additional comments that I would like to see addressed before the manuscript can be accepted.

Title: “Early Holocene Scandinavian foragers on a journey to affluence: Mesolithic fish

exploitation, seasonal abundance and storage investigated through strontium isotope

ratios by laser ablation (LA‐MC-ICP‐MS)”. Rereading the manuscript I don’t see the direct connection or measurements of “affluence”, or quantitative estimates on “abundance”. I suggest rethinking the title with the final manuscript in mind. Its really about fish exploitation using seasonal migratory species.

Major comments

- The results section (baseline data) includes a discussion of the results and provides context. I suggest splitting this into results and discussion and only present the new data obtained in this study in the results. Make everything baseline related a single result section and compile the new baseline Sr isotope ratios into a table (maybe adding lat/lon?, or other geographic identifiers).

- I suggest to combine Figure 5 and Figure 6

- Figure 7 could be a point or a boxplot showing the Sr isotope ranges for all considered units (so published data + the new data from this study).

- What extra information do Figure 9 and 10 provide that is not conveyed in Figure 8? Also double check the colors between figures and legend and maybe avoid red and green together.

- Figure 11, add jittered datapoints to these boxplots (maybe with a grey outline)

- The results of the fish teeth also mix results and discussion which should be clearly separated. If citing literature (other than data sources) it probably should be in the discussion section

- How are the concluding remarks different than the discussion? I first thought of them like a summary but for that purpose that section is too long. Maybe streamline it and move some of the details into the discussion?

At 12 figures + a large supplement the paper is very long when it really doesn’t need to be. The story is relatively simple (and very interesting!) but gets diluted by all the extra details that don’t contribute to the major research questions which if I understand correctly are simply (1) are the fish local (to the lake) or migratory and does this differ between the pit vs non-pit. I strongly suggest to remove anything that doesn’t directly relate to these questions.

Figures in general: Provide clear x and y axis labels (on the outside of the axis not in the plot), report Sr isotopes to the same decimal value, and use a period and not a comma (so 0.70198 instead of 0,70198). I again suggest to redo the figures in R, python, matlab, or similar. This is a really interesting paper but the figures are a major drawback and could be easily improved through use of an alternative to excel software. Also consider color choices as a lot of people may have problems differentiating the different greens and reds.

Minor comments

Abstract

Line 14: “We have analysed 87Sr/86Sr strontium isotope ratios..” reword to “We analysed strontium isotope ratios (87Sr/86Sr)…”

Line 17: “centres of origin”, do you mean “regions of origin”?

Line 17:” Our results suggest”, replace with “We find”

Line 18: Not everyone will know that roach are a cyprinid. Maybe you can reword and say…”the most commonly fermented cyprinid, roach,..” and then in the next sentence add “This is in contrast to the other cyprinidis…”

Line 23: remove “show low, non-local, Sr ratios 23 and” you don’t give the Sr isotope ratios for the other fish

Line 167: replace “its” with “their”

Line 170: Space missing between “Fig1Sand”

Line 191:” Mobility and provenance studies is” it should be “are”

Line 198: That’s not correct. The half-life doesn’t influence if the ratio would be altered when moving from the minerals in the rock to the soil and watershed. The long half-life just means that for archaeological studies no additional Rb decay needs to be accounted for.

Line 205: Remove “of the bioavailable baseline”

Line 260: Replace “over” with “for”

Line 268 (and rest of this paragraph): Remove the “(see Results)”

Line 334: Maybe mention that enamel/dentine mix was excluded

Line 345: I don’t think organics cause larger fractionations. But interferences makes sense

Line 528: If you use the word significant you need to provide a statistical test. Maybe reword?

Line 533. Period missing

Line 558: The first paragraph should pickup where the introduction left off and summarize the results. The following paragraphs can then provide the detailed discussion of the results and the context. Lin 602 might be a good start for the discussion followed by :” We found…”

Supplement:

- The color red often is used to indicate erroneous data. Maybe pick a different color to represent the enamel only value.

- Match font type, heading colors, and general styles to the plosone format.

Reviewer #2: This revised manuscript takes account of the reviewer comments and is substantially improved over the original submission. I see no reason it should not now be published. Alex Pryor.

7. PLOS authors have the option to publish the peer review history of their article (what does this mean?). If published, this will include your full peer review and any attached files.

Reviewer #1: **Yes: **Malte Willmes

Reviewer #2: **Yes: **Alexander J.E. Pryor

---

## [Author Response · Author response to Decision Letter 1]

22 Dec 2020

Answers to reviewer comments have been colour coded and provided in attached file

---

## [Editor Report · Decision Letter 2]

26 Dec 2020

Early Holocene Scandinavian foragers on a journey to affluence: Mesolithic fish exploitation, seasonal abundance and storage investigated through strontium isotope ratios by laser ablation (LA‐MC-ICP‐MS)

PONE-D-20-15007R2

Dear Dr. Boethius,

We’re pleased to inform you that your manuscript has been judged scientifically suitable for publication and will be formally accepted for publication once it meets all outstanding technical requirements.

Kind regards,

Dorothée Drucker

Academic Editor

PLOS ONE
---

## [Editor Report · Acceptance letter]

30 Dec 2020

PONE-D-20-15007R2 

Early Holocene Scandinavian foragers on a journey to affluence: Mesolithic fish exploitation, seasonal abundance and storage investigated through strontium isotope ratios by laser ablation (LA‐MC-ICP‐MS) 

Dear Dr. Boethius:

I'm pleased to inform you that your manuscript has been deemed suitable for publication in PLOS ONE. Congratulations! Your manuscript is now with our production department. 

Kind regards, 

on behalf of

Dr. Dorothée Drucker 

Academic Editor

PLOS ONE